



# Time dependent source apportionment of submicron organic aerosol for a rural site in an alpine valley using a rolling PMF window

Gang Chen[1*], Yulia Sosedova[1*], Francesco Canonaco[1,2], Roman Fröhlich[1], Anna Tobler[1,2], Athanasia Vlachou[1], Kaspar R. Daellenbach[1], Carlo Bozzetti[2], Christoph Hueglin[3], Peter Graf[3], Urs Baltensperger[1], Jay G. Slowik[1], Imad El Haddad[1], and André S.H. Prévôt[1**]

[1]Laboratory of Atmospheric Chemistry, Paul Scherrer Institute, CH-5232 Villigen PSI, Switzerland

[2]Datalystica Ltd., Park innovAARE, CH-5234 Villigen, Switzerland

[3]Empa, Swiss Federal Laboratories for Materials Science and Technology, Laboratory for Air Pollution and Environmental Technology, CH-8600 Dübendorf, Switzerland

*G.C. and Y.S. contributed equally to this manuscript*

**Correspondence to:* André S. H. Prévôt (andre.prevot@psi.ch)




**Abstract**
We have collected one year of aerosol chemical speciation monitor (ACSM) data in Magadino,
a village located in the south of the Swiss Alpine region, which is one of the most polluted
areas in Switzerland. We analysed the mass spectra of organic aerosol (OA) by positive matrix
factorization (PMF) using Source Finder Professional (SoFi Pro) to retrieve the origins of OA.
Therein, we deployed the rolling algorithm to account for the temporal changes of the source
profiles, which is closer to the real world. As the first ever application of rolling PMF analysis
for a rural cite, we resolved two primary OA factors (traffic-related hydrocarbon-like OA
(HOA) and biomass burning OA (BBOA)), one local OA (LOA) factor, a less oxidized
oxygenated OA (LO-OOA) factor, and a more oxidized oxygenated OA (MO-OOA) factor.
HOA showed stable contributions to the total OA through the whole year ranging from 8.1–
10.1%, while the contribution of BBOA showed a clear seasonal variation with a range of 8.3–
27.4% (highest during winter, lowest during summer) and a yearly average of 17.1%. The OOA
was represented by two factors (LO-OOA and MO-OOA) throughout the year. OOA
contributed 71.6% of the OA mass, varying from 62.5% (in winter) to 78% (in spring and
summer). The uncertainties ($\sigma$) for the modelled OA factors (i.e., rotational uncertainty and
statistical variability of the sources) varied from ±4% (LOA) to a maximum of ± 40% (LO-
OOA). Considering the fact that BBOA and LO-OOA (showing influences of biomass burning
in winter) had significant contributions to the total OA mass, we suggest a reduction and control
of the residential heating as a mitigation strategy for better air quality and lower PM levels in
this region. In Appendix A, we conducted a head-to-head comparison between the conventional
seasonal PMF analysis and the rolling mechanism. It showed similar or slightly improved
results in terms of mass concentrations, correlations with external tracers and factor profiles of
the constrained POA factors. The rolling results show smaller scaled residuals and enhanced
correlations between OOA factors and corresponding inorganic salts than those of the seasonal



solutions, was most likely because the rolling PMF analysis can capture the temporal variations
of the oxidation processes for OOA sources. Specifically, the time dependent factor profiles of
MO-OOA and LO-OOA can well explain the temporal viabilities of two main ions for OOA
factors, $m/z$ 44 ($CO_2^+$) and $m/z$ 43 (mostly $C_2H_3O^+$). This rolling PMF analysis therefore
provides a more realistic source apportionment (SA) solution, with time-dependent OA sources.
The rolling results show also good agreement with offline Aerodyne aerosol mass spectrometer
(AMS) SA results from filter samples, except for winter. This is likely because the online
measurement is capable of capturing the fast oxidation processes of biomass burning sources.
This study demonstrates the strengths of the rolling mechanism and provides a comprehensive
criterion list for ACSM users to obtain reproducible SA results and is a role model for similar
analyses of such world-wide available data.

## 1   Introduction

Atmospheric particulate matter (PM) affects human health and climate. In particular, it
influences the radiative balance (IPCC, 2014; von Schneidemesser et al., 2015), reduces
visibility (Chow et al., 2002; Horvath, 1993), and negatively affects human health by triggering
respiratory and cardiovascular diseases and allergies (Daellenbach et al., 2020; Dockery and
Pope, 1994; Mauderly and Chow, 2008; Monn, 2001; Pope and Dockery, 2006; von
Schneidemesser et al., 2015). Fine PM exposure strongly correlates with the global mortality
rate. Lelieveld et al. (2015) estimated that outdoor air pollution, mostly $PM_{2.5}$ (PM with an
aerodynamic diameter smaller than 2.5 µm) causes 3.3 million premature deaths per year
worldwide. Despite of this correlation, different aerosol sources may have strongly different
effects on health (Daellenbach et al., 2020). Thus, both climate and health effects are affected
by particle chemical composition, which is related to emission sources of primary particles and



precursor gases for secondary aerosol (IPCC, 2014; Jacobson et al., 2000; Jacobson, 2001;
Lelieveld et al., 2015; Ramanathan et al., 2005).
Organic aerosol (OA) constitutes 20–90% fine PM (Jimenez et al., 2009; Murphy et al., 2006;
Zhang et al., 2007), and contains millions of chemical compounds. Since OA is subject of an
extremely complex mixture of chemical constituents, with highly dynamic spatial and temporal
(seasonal, diurnal, etc.) variability of directly emitted particles and gas-phase precursors and a
complex chemical processing in the atmosphere, elucidation of the chemical composition and
physical properties of OA remains challenging. Identification and quantification of OA sources
with a sophisticated interpolation of both spatial and temporal variabilities are essential for a
development of effective mitigation strategies for air pollution and a better assessment of the
aerosol effect on both health and climate.
OA source apportionment (SA) and PM composition has been studied extensively using the
Aerodyne aerosol mass spectrometer (AMS) (Canagaratna et al., 2007). However, due to the
complexity of the AMS measurements and their high operational expenses, AMS campaigns
are often limited to short time periods of a few weeks to months. The aerosol chemical
speciation monitor (ACSM) allows for unattended long-term observation (>1 year) of non-
refractory aerosol particles (Ng et al., 2011a; Fröhlich et al., 2013). It makes it possible to
investigate also the long-term temporal variations of OA sources, which is crucial for
policymakers to introduce or validate aerosol-related environmental policies.
Positive matrix factorization (PMF) has been used in various studies for SA of OA (Aiken et
al., 2009; Hildebrandt et al., 2011; Lanz et al., 2007; Mohr et al., 2012; Schurman et al., 2015;
Zhang et al., 2011). The multilinear engine (ME-2) implementation of PMF (Paatero, 1999)
improves model performance by allowing the use of *a priori* information (constraints on source
profiles and/or time series) to direct the model towards environmentally meaningful solutions



(Canonaco et al., 2013; Crippa et al., 2014; Fröhlich et al., 2015; Lanz et al., 2008; Ripoll et
al., 2015). For long-term data (one year or more) with high time resolution, the composition of
a given source could change considerably due to the meteorological and seasonal variabilities.
However, a major limitation of PMF is the assumption of static factor profiles, such that it fails
to respond to these temporal changes. Therefore, long-term chemically speciated data have
been evaluated monthly or seasonally (Bressi et al., 2016; Canonaco et al., 2015; Minguillón
et al., 2015; Petit et al., 2014; Reyes-Villegas et al., 2016; Ripoll et al., 2015) to at least take
the seasonal variations into account. To improve analysis of long-term ACSM datasets, a novel
approach that utilizes PMF analysis on a smaller time rolling window was first proposed by
Parworth et al. (2015) and further refined using ME-2 by Canonaco et al. (2020). The short
length of the rolling PMF window allows the PMF model to take the temporal variations of the
source profiles into account (e.g., biogenic versus domestic burning influences on oxygenated
organic aerosol (OOA)), which normally provides a better separation between OA factors. In
addition, using this technique together with bootstrap resampling and a random $a$-value
approach allows users to assess the statistical and rotational uncertainties of the PMF results
(Canonaco et al., 2020; Tobler et al., 2020).
In this work, we conducted a one year ACSM measurement from September 2013 to October
2014 in Magadino, located in an alpine valley in southern Switzerland. We present a
comprehensive analysis of the ACSM dataset measured in Magadino using a novel PMF
technique, the "rolling PMF". In addition, we also compare the results of the rolling PMF with
the source apportionment of offline AMS filter samples (Vlachou et al., 2018) and conventional
seasonal PMF analysis.


## 2 Methodology

### 2.1 Sampling site

Magadino is in a Swiss alpine valley (46°90'37'' N, 85°60'2'' E, 204 m.a.s.l.), where the sampling site located. This site belongs to the Swiss National Air Pollution Monitoring Network (NABEL, https://www.empa.ch/web/s503/nabel). It is around 1.4 km away from the local train station, Cadenazzo, around 7 km away from the Locarno Airport, and nearly 8 km away from the Lake Maggiore. This station is surrounded by agricultural fields within a rural area, which is considered as a rural background site. It can be potentially affected by domestic wood burning, adjacent agricultural activity and transit traffic through the valley. The site topography favours quite high PM levels due to stagnant meteorological conditions or boundary layer inversions, especially in winter. The annual average $PM_{10}$ concentration in Magadino exceeded the annual average $PM_{10}$ limit value for Switzerland (20 $\mu g \cdot m^{-3}$) for five years out of the period 2007–2016 (Meteotest, 2017; The Swiss Federal Council, 2018).

### 2.2 ACSM measurements

In this study, chemical composition and mass loadings of non-refractory constituents of ambient submicron aerosol particles ($NR\text{-}PM_1$) were measured by an Aerodyne quadrupole ACSM (Ng et al., 2011a). The ACSM uses the same sampling and detection technology as the AMS but is simplified and designated for long-term monitoring applications by reducing maintenance frequency, at the cost of lower sensitivity, restriction to integer mass resolution, and no size measurement. Same as for the AMS, sampled submicron particles enter the instrument through a critical orifice (100 $\mu m$ I.D.) at a flow rate of 1.4 $cm^3$ $s^{-1}$ (at 20 °C and 1 atm). The sampling flow will pass either through a particle filter or directly into the system using an automated 3-way switching valve, that is switched every ~30 s. The sampled particles are focused by an aerodynamic lens into a narrow beam and impact on a tungsten surface of around 600 °C, where the non-refractory particles vaporize and are subsequently ionized by an





electron impact source (70 eV). The resulting ions are detected by a quadrupole mass-
spectrometer up to a mass to charge ratio $m/z = 148$ Th. The particle mass spectrum is
represented by the difference of the total ambient air signal and the particle-free signal.
The quantification of ACSM data requires an estimation of the fraction of NR-PM$_1$ that
bounces off the oven without being vaporized and therefore is not detected (Canagaratna et al.,
2007; Matthew et al., 2008). A collection efficiency (CE) factor is typically introduced to
correct for particle bounce, which depends on the particulate water content (Matthew et al.,
2008), ammonium nitrate mass fraction (ANMF) and acidity (Middlebrook et al., 2012). To
eliminate humidity effects on CE, a Nafion membrane dryer (Perma Pure MD) was installed
on the sampling inlet. In this study, we compared both, a constant CE of 0.45 and a time-
dependent CE correction suggested by Middlebrook et al., (2012). It showed that data corrected
with a constant CE had a better correlation and slope closer to 1 when comparing with the
chromatographic $SO_4^{2-}$, $NO_3^-$, and $Cl^-$ anions (Fig. S1a). In addition, as more than 93.5% data
have an ANMF smaller than 0.4, only 6.5% of data would be impacted by a time-dependent
CE correction, therefore, the ammonium nitrate particles doesn't have significant effects on
CE for this dataset. Overall, this dataset agrees with external TEOM measurement of both
PM$_{2.5}$ and PM$_{10}$ daily mass concentrations as shown in Fig S1c with a constant CE value.
The ACSM filament burnt out on 14 April, 2014. This was addressed by switching to the
backup filament already installed within the instrument (no venting required). Calibration of
the relative ionization efficiencies (RIE) of particulate nitrate, sulphate, and ammonium was
conducted using size-selected (300 nm) pure $NH_4NO_3$ and pure $(NH_4)_2SO_4$ particles.
Calibrations of the relative ionisation efficiency (RIE), $m/z$ scale, and the sampling flow was
performed every 2 months. In this study, we used the averaged RIEs for nitrate, sulphate, and
ammonium, the exact values are shown in Fig S1.





## 2.3   Complementary measurements
Meteorological data, including temperature, precipitation, wind speed, wind direction, and
solar radiation are monitored at the NABEL station. In addition, concentrations of trace gases
($SO_2$, $O_3$, $NO_x$), equivalent black carbon (eBC), and $PM_{10}$ were measured with a time resolution
of 10 minutes. We used an aethalometer (AE 31 model by Magee Scientific Inc.) to measure
eBC concentrations. Therefore, we conducted SA of eBC by following Zotter et al. (2017)
using Ångstrom exponents for eBC from traffic $\alpha_{tr} = 0.9$ and wood burning $\alpha_{wb} = 1.68$.
More details about eBC source apportionment are provided in Section 1 of the SI.
## 2.4   Preparation of the data and error matrices for PMF
In this study, we used acsm_local_1610 software (Aerodyne Research Inc.) to prepare the PMF
input matrix. In total, this dataset includes 19'708 time points and 67 ions. Of these, $CO_2^+$-
related variables ($I_{O^+}$ ($m/z = 16$), $I_{HO^+}$ ($m/z = 17$), and $I_{H2O^+}$ ($m/z = 18$)) were excluded from the
spectral matrix prior to a PMF analysis. They are reinserted into the OA factor mass spectra
after the PMF analysis using the ratio from the fragmentation table (Allan et al., 2004); the
factor concentrations are likewise adjusted. The measurement error matrix was calculated
according to Allan et al. (2003, 2004), with a minimum error considered for the uncertainty of
all variables in the data matrix as in Ulbrich et al. (2009). Following the recommendations in
Paatero and Hopke (2003) and Ulbrich et al. (2009), the measurement uncertainty for variables
($m/z$) with a signal-to-noise ratio (S/N) < 2 (weak variables) and S/N < 0.2 (bad variables) were
increased by a factor of 2 and 10, respectively. In total, 27 weak ACSM variables were down-
weighted. Additionally, $m/z$ 12 and 13 were not considered during the PMF analyses, due to
being noisy and their overall negative signal. Moreover, $m/z$ 15 is not only very noisy (S/N =
0.09), but may be also affected by high biases due to potential interference with air signals.


## 2.5   Factor analysis of the organic mass spectra
PMF has been demonstrated to be a useful tool to retrieve the sources of measured organic
aerosol mass spectra with a bilinear factor model (Paatero and Tapper, 1994; Ulbrich et al.,

184    2009):


$$x_{ij} = \sum_{k=1}^{p} g_{ik} \times f_{kj} + e_{ij}$$
(1)


where $x_{ij}$ is the mass concentration of the $j^{th}$ mass spectral variable in the time point $i^{th}$; $g_{ik}$
is the contribution of the $k^{th}$ factor in the $i^{th}$ time point; $f_{kj}$ is the concentration of the
$j^{th}$ mass spectral variable in the $k^{th}$ factor; and $e_{ij}$ is the residual of $j^{th}$ variable of the mass
spectra in $i^{th}$ time point. The superscript, $p$ represents the number of factors, which is
determined by the user. The cost function of PMF uses least squares algorithm by iteratively
minimizing the following quantity Q:

$$Q = \sum_{i=1}^{n} \sum_{j=1}^{m} (\frac{e_{ij}}{\sigma_{ij}})^2$$
(2)


where $\sigma_{ij}$ is an element in the $n \times m$ matrix of the measurement uncertainties, which
corresponds point-by-point to $x_{ij}$. In addition, we normalized quantity $\frac{Q}{Q_{exp}}$ as a mathematical
metric during PMF analysis, where the $Q_{exp}$ is:



$$Q_{exp} = (n \times m) - p \times (n + m) \tag{3}$$


The $\frac{Q}{Q_{exp}}$ supports the user to determine the number of factors required for the model by
investigating the effects on this quantity of adding/removing a factor. However, PMF itself
suffers from rotational ambiguity because of the fact that the object function, Q does not
provide unique solutions, that is when $\mathbf{G} \cdot \mathbf{F} = \mathbf{G} \cdot \mathbf{T} \cdot \mathbf{T^{-1}} \cdot \mathbf{F}$, PMF provides a similar value of
Q but very different solutions (rotated matrix $\overline{\mathbf{G}} = \mathbf{G} \cdot \mathbf{T}$ (rotated factor time series) and $\overline{\mathbf{F}} =$
$\mathbf{T^{-1}} \cdot \mathbf{F}$ (rotated factor profiles)). Only one of or even none of these rotated solutions may be
atmospherically relevant. The ME-2 solver (Paatero, 1999) enables theoretically full rotational
control over the factor solutions, which is implanted here by imposing constraints via the *a-*
value approach on one or more elements of $\mathbf{F}$ and/or $\mathbf{G}$ (Paatero and Hopke, 2009). The *a*-value
(ranging from 0 to 1) determines how much the resulting factor ($f_{j,solution}$) or time series
($g_{j,solution}$) can vary from the input reference factor ($f_{j,reference}$) or time series ($g_{j,reference}$)
as shown in Eq. 4a and 4b:

$$f_{j,solution} = f_{j,reference} \pm a \cdot f_{j,reference} \tag{4a}$$

$$g_{j,solution} = g_{j,reference} \pm a \cdot g_{j,reference} \tag{4b}$$


Previous work using *a*-values has shown to efficiently retrieve environmentally reasonable
PMF solutions. This is due to the presence of legitimate *a priori* constraints which decrease the
degree of rotational ambiguity (Canonaco et al., 2013, 2020; Crippa et al., 2014; Lanz et al.,
2008). Here we configured the ME-2 solver and analysed PMF results using SoFi (Source



Finder, Datalystica Ltd., Villigen, Switzerland) Pro 6.D interface (Canonaco et al., 2013, 2020),
developed within the IGOR Pro software (WaveMetrics Inc., Lake Oswego, OR, USA).
Running PMF over the long-term ACSM datasets assumes that the OA source profiles are static
within this time window. This can lead to large errors, since OA chemical fingerprints are
expected to vary over time (Paatero et al., 2014). For example, Canonaco et al. (2015) showed
that the variability of summer and winter OOA cannot be accurately represented by a single
pair of OOA profiles. A common way to reduce the model uncertainty arising from this source
is to choose a proper number of OA factors (Sug Park et al., 2000), and then perform a PMF
analysis on a subset of measurements to capture temporal features of OA chemical fingerprints.
Such characterization of OA sources on a seasonal basis has been demonstrated in a number of
studies (Crippa et al., 2014; Lanz et al., 2008; Minguillón et al., 2015; Petit et al., 2014; Ripoll
et al., 2015; Zhang et al., 2019).
## 2.6  Rolling PMF analysis with ME-2
In this study, we performed PMF runs with *a priori* constraints (factor profiles) retrieved from
seasonal bootstrap analysis (Section 2.2 in the SI) on a small and rolling window (i.e., 1, 7, 14,
and 28 days) that could move across the entire dataset with a step of one day (Canonaco et al.,
2020; Parworth et al., 2015). In addition, we used the bootstrap re-sampling strategy, which
can randomly choose a subset of the original matrix and replicate some of the rows/columns to
create a new same-size matrix (Efron, 1979). Here, we combined this rolling PMF analysis
with the bootstrap strategy and random *a*-values for constrained factor profiles to estimate the
statistical and rotational uncertainties of this PMF analysis. More details of this novel technique
is found in Canonaco et al. (2020).



### 2.6.1 Window settings


In order to retrieve appropriate constraints, we performed PMF *pre-tests* and bootstrap analysis
for different seasons. More details of the steps, settings of these analysis can be found in Section
2 of the SI. Here, we constrained primary OA factor profiles (hydrocarbon-like OA factor
(HOA) and biomass burning OA (BBOA)) as well as the factor profile of a local factor (LOA)
using the *a*-value technique in the rolling PMF analysis. The reference profiles of HOA and
BBOA were from the winter bootstrapped PMF solution (Dec, Jan, and Feb) as shown in Fig.
S6. With a higher contribution of the biomass burning trace ion *m/z* 60 in the winter, we expect
a more representative and robust BBOA profile from the winter solution than from other
seasons. The LOA profile was retrieved from the summer bootstrapped PMF solution (Jun, Jul,
and Aug) (Fig. S6). To allow the factor profile to adapt itself over time, a random *a*-value
within a range of 0.4 with a step of 0.1 is applied for HOA and BBOA. Canonaco et al. (2020)
suggested that an upper *a*-value of 0.4 is sufficient to cover the temporal variation of OA source
profiles. Moreover, due to the uniqueness of the LOA chemical profile, it is tightly constrained
with a constant *a*-value of 0.05. The LOA factor appeared only after the filament had been
changed (14 April, 2014), and its mass spectrum is dominated by nitrogen-containing
fragments (at *m/z* 58, 84, and 98). The instrument setup thus influenced strongly the sensitivity
of these components (likely due to influences of surface ionization). Therefore, this factor was
considered in the PMF analysis, but no further interpretation of its potential source will be
covered in this manuscript.
In total, we constrained HOA and BBOA factors with random *a*-value (0–0.4, with a step of
0.1), and an exact *a*-value (0.05) for LOA factor in the rolling PMF analysis. There are 25
(*N*=5×5) possible *a*-value combinations within an individual rolling window. Therefore, 50
PMF iterations for each time window are sufficient to cover all possibilities of the *a*-value
combinations. With the rolling window of 50 repeats, each data point (except the data within



the first and last time window) will actually have many PMF iterations (i.e., $N$=length of the
window$\times$50), where bootstrap resampling and random combinations of constraints is
performed. This allows to estimate the statistical and rotational uncertainties of the PMF factors
(Canonaco et al., 2020). To find the optimum length of the time windows, we tested four
different lengths of the time windows (N=1, 7, 14, 28) using the same approaches as in
Canonaco et al. (2020). We determined the optimum length of the time window based on the
number of missing data points (un-modelled data due to the selection based on the criteria)
while applying the same thresholds for the same criteria.
### 2.6.2  Criteria settings
Performing a rolling analysis for a one-year data with 50 repeats per window requires several
tens of thousands of PMF runs. Manual inspection of all PMF runs is impractical and therefore
was replaced by monitoring user-defined criterion scores (Canonaco et al., 2020). In this study,
$R^2$ values of the time series of modelled HOA vs $NO_x$ and $eBC_{tr}$ were used for HOA. The
BBOA factor was inspected using the variation of $m/z$=60 explained by BBOA (Table S1). For
these time series based criteria, (criterion 1 to criterion 3 in Table S1), we deployed student t-
test to minimize subjective judgment while determining the thresholds (more discussions in
Section 2.3 of the SI).
Typically, OOA factors are dominated by the signals of $f_{43}$ ($C_2H_3O^+$ at $m/z = 43$) and $f_{44}$ ($CO_2^+$
at $m/z = 44$) that correspond to the less and more oxygenated ion fragments (Canonaco et al.,
2015; Ng et al., 2010), where $f$ is the fraction of a variable, $i.e.$ the intensity $I_{m/z}$ normalized by
the sum of the intensities of all organic $m/z$ variables. In this study, we were able to retrieve
two OOA factors (i.e., more oxidized OOA (MO-OOA) and less oxidized OOA (LO-OOA))
for the whole year, while MO-OOA can be at either at 4[th] or 5[th] position because there are two
unconstrained factors. Thus, we used the $f_{44}$ for the 4[th] factor to sort the unconstrained OOA



factors to ensure MO-OOA and LO-OOA sitting on the $4^{th}$ and $5^{th}$ position, respectively. The
details of the sorting scheme can be found in Canonaco et al. (2020). At the same time, we also
monitored the $f_{43}$ in LO-OOA and $f_{44}$ in MO-OOA to make sure they are not zero. With this set
of criteria, we were able to only select "good" (atmospherically relevant) PMF runs before
averaging.
**3   Results and discussion**
3.1   Overview of $PM_1$ sources in Magadino
Considering that the major part of eBC is within $PM_1$ (Schwarz et al., 2013), we added eBC to
the total NR-$PM_1$ from the ACSM to perform a mass closure analysis with $PM_{2.5}$/$PM_{10}$ from
filters. The gravimetric $PM_{2.5}$ and $PM_{10}$ show a high correlation with total estimated $PM_1$ (NR-
$PM_1$ +eBC) (Fig. S1c). The slopes of the linear fits ($\pm$ 1 standard deviation) are $1.62 \pm 0.05$ ($R^2$
$= 0.81$, $N$=79) for $PM_{2.5}$ *vs.* $PM_1$ and $1.84 \pm 0.03$ ($R^2 = 0.67$, $N$=335) for $PM_{10}$ *vs.* $PM_1$. This
means that the estimated $PM_1$ comprised 62% and 54% of the $PM_{2.5}$ and $PM_{10}$ mass,
respectively. The daily averages of the inorganic species concentrations measured by the
ACSM and those measured on the filters by chromatography show a high correlation, with $R^2$
$= 0.83$ for $SO_4^{2-}$, $R^2 = 0.82$ for $NO_3^-$ and $R^2 = 0.50$ for $Cl^-$, with slopes close to 1 (Fig. S1a).
The 2-week average of total ammonium and total nitrate measured by offline AMS technique
agree rather well with the ACSM ammonium ($R^2 = 0.47$) and nitrate ($R^2 = 0.79$), as shown in
the plots in Fig. S1b. The ion balance of particulate ammonium, sulphate and nitrate measured
by the ACSM showed that the measured aerosol particles were mostly neutral.
The daily average $PM_1$ components are shown in Fig. 1a, with the annual average $PM_1$
concentration (including eBC) for the period from September 2013 to October 2014 equal to
10.2 $\mu$g m$^{-3}$. In winter, the average $PM_1$ concentration was highest (13.8 $\mu$g·m$^{-3}$), with OA





contributing 54% to the total $PM_1$ mass. In summer, the average $PM_1$ mass concentration was
below 10 µg·m$^{-3}$, but the relative contribution of the OA fraction increased to 62%.
Seasonally averaged diurnal cycles of NR-$PM_1$ and of eBC are displayed in Fig. 2. In this study,
all the data is based on local time (Central European Time). In fall, spring and summer, the
diurnals of these pollutants seem to be mainly affected by the development of the BLH, most
of the species show similar diurnal trends for these three seasons. In addition, summer has the
highest sulphate concentration, due to the enhanced photochemical production. In winter, air
pollutants accumulated during evening and night due to the thermal inversion. In general, eBC
and organics have higher levels due to enhanced biomass burning emissions and lower
boundary layer height (BLH). We observed distinct midday peaks of organics, sulphate, nitrate,
ammonium, chloride, and $NO_x$ in the winter. Magadino experienced a series of windless, cold,
but sunny periods from December 2013 to January 2014, including such sharp peaks (Fig. S3a).
It is interpreted to be due to advection within the shallow boundary layer due to the fact that
both primary and secondary pollutants increased simultaneously. Local winds were very low
near the ground but likely locally and regionally induced orography influenced winds including
vertical diffusion processes were initiated during these times that are difficult track without
spatially distributed measurements. . Such phenomena were not observed during cloudy, cold,
and windless days (Fig. S3b) without thermally induced meteorological processes. Unlike other
seasons, the dilution process due to vertical mixing happened only after noon time due to strong
inversions during the night and late irradiation of the valley surface in winter.
## 3.2   Seasonal PMF *Pre-tests*
The automated rolling PMF analysis requires the knowledge of the reference profiles as well
as the number of factors. In this section, we present how number of factors were determined
based on seasonal PMF *pre-tests*. Initially, unconstrained PMF (3 to 6 factors) was performed



separately for the different seasons by following the SA guidelines provided by Crippa et al.
(2014). Typically, the HOA profile is characterized by a high contribution of alkyl fragments
(*e.g.* $m/z$ =43, $m/z$ =57) and the corresponding alkenyl carbo cations (*e.g.* $m/z$ = 41, $m/z$ = 55),
and the factor profile is relatively consistent over time and different locations. The BBOA
profile exhibits significant signals at $m/z$ = 60 and $m/z$ =73, which are well-known fragments,
arising from fragmentation of anhydrous sugars present in biomass-related emissions (Alfarra
et al., 2007). For the unconstrained PMF runs, the HOA profile is present throughout the whole
year, while the BBOA profile exists for all seasons except in summer. However, as shown in
Fig. S4, the measured fraction of $m/z$ = 60 during summer was above the background level of
biomass burning-related air masses, 0.3% ±0.06% (Aiken et al., 2009; Cubison et al., 2011;
DeCarlo et al., 2008). In addition, the scaled residual at $m/z$ = 60 was decreased when a BBOA
factor profile was constrained. Thus, we decided to constrain the BBOA factor for all seasons
to potentially capture some local events, such as agricultural and open fires in summer.
No evidence for the presence of a cooking-related OA (COA) factor was found based on the
seasonal pre-analysis of the key fragments ($m/z$ 55 and $m/z$ 57). It shows no difference in the
slope of the absolute mass concentration of $m/z$ 55 vs $m/z$ 57 for different hours of the day (Fig
S5a), while different seasons show different slopes (Fig S5b). Therefore, a COA factor was not
considered in the PMF model. Moreover, a rapid increase of the measured fraction of $m/z$ = 58,
84, and 98 together with $m/z$ 39 (potassium signal) was observed after a filament exchange on
14 April, 2014. It is likely that the ACSM's sensitivity towards those ions was changed by the
filament exchange. Also, this LOA factor was present for spring, summer, and autumn in 2014
in unconstrained PMF runs all the time after the filament change. Therefore, we kept this factor
for these three seasons.



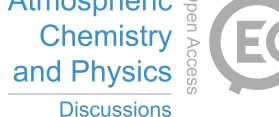

For the factor(s) with secondary origin, PMF models with different number of factors (3–6)
were tested to assess if the oxygenated OA (OOA) factor (with a high contribution of $m/z$ 44
that is likely dominated by the $CO_2^+$ ion, derived from decomposition of carboxylic acids
(Duplissy et al., 2011)) is separable without mixing with primary organic aerosol (POA) factors
(Fig. S6). We conducted these tests independently for different seasons (autumn 2013, winter,
spring, summer, autumn 2014).
We analysed winter data first by constraining a HOA factor profile (Crippa et al., 2013) with a
tight $a$-value of 0.05. The 3-factor solution (with one OOA factor) showed similarly good
agreement of HOA and BBOA with the external tracers ($NO_x$, $eBC_{tr}$, $eBC_{wb}$) as the 4-factor
solution (with two OOA factors). However, the scaled residual of $m/z$ 60 was reduced for the
solution with two OOA factors. Moreover, the solution with one OOA factor was not sufficient
to explain the variabilities of measured $f_{44}$ vs $f_{43}$ (excluding the primary organic aerosol (POA)
factors). For 5- and 6-factor solutions the BBOA and LO-OOA factors started to split.
Eventually, we selected the 4-factor solution (HOA, BBOA, MO-OOA, LO-OOA) as the best
representation of the winter data.
After the bootstrap seasonal PMF runs of winter data (details in Section 2 of the SI), we
extracted the HOA and BBOA profiles to use them as the reference factor profiles (Fig. S6) for
the *pre-tests* of other seasons. For the spring, summer, and autumn seasons, 3- to 6-factor PMF
solutions were modelled separately for each season by constraining the HOA ($a$-value=0.1)
and BBOA ($a$-value=0.3) profiles. For the 3-factor solution, we observed an OOA factor with
some signals at $m/z$ 58, 84, and 98 which we could not relate to a specific source or process.
Also the scaled residuals of variables showed significant levels for these three ions. When we
increased the number of OA factors from 3 to 4, a factor dominated by $m/z$ 58, 84, and 98
emerged, which we named local organic aerosol (LOA). However, the OOA factor still showed





slight signals at *m/z* 58, 84, and 98. An increase in the number of factors from 4 to 5 did not
only result in a decrease in $\frac{Q}{Q_{exp}}$, but also in "clean" OOA factors without mixing with the LOA
factor. A further increase in the number of factors did not change $\frac{Q}{Q_{exp}}$ substantially ($< 1\%$),
and the sixth factor was a mathematical split of the LOA factor with *m/z* 58 as the dominating
variable. Thus, the 5-factor PMF model was chosen as the most appropriate for the spring,
summer, and autumn 2014. Note that we did not add the LOA factor for the autumn season in
2013 since it appeared only after the filament exchange on 14 April, 2014. This LOA factor
was included while running PMF because of the rapid drop of the $\frac{Q}{Q_{exp}}$ from 4 to 5 factors in
the PMF model, but the source of this factor will not be discussed in the manuscript.
## 3.3   Full year rolling PMF analysis
Here we present the optimized time window size (14 days) (details of the time window
optimization are given in Section 4 of the SI and Fig S10). In total, we considered 53.4% of
the PMF runs (11087 out of 20750) with only 11 non-modelled data points. The results of the
full-year PMF analysis of the 30-min resolved ACSM data are summarized in Fig. 3. The
relative contributions of the OA factors are in addition shown in Fig. 3b. The primary traffic
related HOA had very little variation (seasonal averages between 8.1 and 10.1%) throughout
the year (Fig. 4). In contrast, BBOA showed a distinct yearly cycle (8.3–27.4%) with a yearly
averaged contribution of 17.1%. It increased significantly (to 27.4%) in winter which is typical
for Alpine valleys (Szidat et al., 2007). It means that biomass burning was the most important
primary OA source during the cold season in Magadino. The eBC$_{wb}$ showed similar trends as
the BBOA factor time series during the cold seasons (Fig. 3c). The contribution of LOA
remained small before the filament was changed on 14 April, 2014, which is expected because
we could not retrieve this factor in seasonal unconstrained PMF runs before April 2014.





In this study, we retrieved two OOA factors, LO-OOA and MO-OOA. Total OOA (LO-
OOA+MO-OOA) contributed substantially to the total OA mass throughout the whole year
with an average contribution of 71.6% (Fig. 3b; Fig. 4). In general, the contribution of OOA to
the total OA mass did not vary distinctly over the seasons, but reached a maximum of 90.1%
on 12 June, 2014, the day with the highest daily average temperature (30.7 °C).
In this work, we did head-to-head comparisons between the bootstrap seasonal solutions and
the rolling PMF results (see Fig. A1, Fig. A2, Fig. A3, and Table A1 in the Appendix) in terms
of mass concentrations, factor profiles, scaled residuals, and correlations between time series
for each factor and corresponding external tracers. We found consistent factor profiles and
mass concentrations for the constrained factors (i.e., HOA, BBOA, and LOA), while OOA
factors showed quite some differences in both mass concentrations and factor profiles. Rolling
PMF provided slightly better correlations and smaller scaled residuals, therefore, we consider
rolling PMF results to be more environmentally reasonable than those of the seasonal PMF
(more details in Appendix A).
### 3.3.1   Optimized OA factors retrieved from a rolling PMF model
The primary and secondary OA factors retrieved as an annual mean of all optimized PMF
solutions together with their diurnal cycles for all seasons are shown in Fig. 5. Seasonal
variations of the OOA factor profiles are demonstrated in Fig. 7 and further discussed in more
detail in Section 3.3.2. Note that the primary factors (HOA, BBOA, and LOA) were constrained,
where the LOA profile was tightly constrained with an *a*-value of 0.05 due to the uniqueness
of its chemical profile. Therefore, only a small variation was allowed for LOA, while the HOA
and BBOA model profiles varied more due to looser constraints (Fig. S8). HOA and BBOA
have averaged *a*-values of 0.207, and 0.195, respectively. In addition, they both had good
agreement with previous studies (Crippa et al., 2014; Ng et al., 2011b). The probability





distribution function (PDF) of applied $a$-values over time was also investigated (Fig. S8). Most
selected runs chose $a$-values of 0.1–0.3 for HOA and BBOA. The OOA factors show larger
variations in the chemical profiles because these two factors were not constrained due to the
high variability of oxidation processes governing the secondary factors.
Due to extensive residential wood combustion combined with winter inversions, the
concentrations of BBOA and $eBC_{wb}$ were three times higher at night than at midday. As
discussed above, during winter, all of the air pollutants, including all PMF factors peaked
concurrently at 10–11 a.m. (local time) due to development of the mixed boundary layer (light
blue markers in Fig. 2 for total $PM_1$ and Fig. 5b). In summer, an additional local photochemical
production led to an increasing MO-OOA mass during the day (red markers in Fig. 5b),
similarly to the diurnal behaviour of sulphate ($R^2$=0.63). A night-time increase and a daytime
decrease of the LO-OOA mass during spring and summer apparently followed condensation
and re-evaporation cycles of semi-volatile species, similar to the behaviour of ammonium
nitrate. Additionally, nocturnal chemistry of $NO_3/N_2O_5$ radicals could lead to formation of
$HNO_3$ *via* $N_2O_5$ hydrolysis and of organic nitrates *via* oxidation of VOCs (Brown et al., 2004;
Dentener and Crutzen, 1993), thus influencing the diurnal cycles of both particulate nitrate and
LO-OOA (with $R^2 = 0.48$ for spring and $R^2 = 0.36$ for summer).
In Fig. 6, we also present the diurnal cycles of HOA, $eBC_{tr}$ and $NO_x$ with different patterns for
weekdays and weekends. The hourly averages of HOA and $eBC_{tr}$ as well as the $NO_x$ mixing
ratio peak during the morning and evening rush hours over the weekdays, while on the
weekends there is only an evening pollution increase coinciding with the time when people
come back from holidays or night-time leisure activities.





### 3.3.2  $f_{44}/f_{43}$ analysis of secondary OA factors


While $m/z$ 44 is mostly from the fragment of $CO_2^+$, a fingerprint of oxygenated species, $m/z$ 43
can originate from $C_2H_3O^+$ (a fingerprint of semi-volatile species) or $C_3H_7^+$ (a fingerprint of
the primary emissions of hydrocarbon-like species) (Canonaco et al., 2015; Duplissy et al.,
2011; Ng et al., 2010). Thus, $f_{44}$ and $f_{43}$ are often used to identify the oxidation state of the
factors, which is important to differentiate the MO-OOA and LO-OOA factors. Under the
premise that the POA factors and the LOA factor are all well-resolved, it is important to
investigate the relationship between the $m/z$ 44 and $m/z$ 43 signals in the OOA factors to
determine whether or not one/two OOA factors are sufficient to explain the dataset. In addition,
the shapes of the clouds shown in an $f_{44}$ vs $f_{43}$ plot may also include some source-related
information. Figure 7 depicts the relationship between $f_{44}$ and $f_{43}$ of the two modelled OOA
factors for different seasons. The yellow cloud of data points represents the measured $f_{44}$ vs $f_{43}$
after subtracting the $m/z$ 44 and $m/z$ 43 signals contributed by the primary HOA, BBOA and
LOA factors. They are colour coded by the total OA mass concentration (data points with OA
mass concentration below 2 µg·m$^{-3}$ are hidden).
As shown in Fig. 7a, the data points in Sep–Oct (both in 2013 and 2014) were located on the
right side of the triangle presented first by Ng et al. (2010), while the November (2013) data
points were located within the triangle. In addition, the spring and summer data points (Fig. 7c
and Fig. 7d) were all located rather on the right side of the triangle, but the winter points lied
within the triangle (Fig. 7b). The data points located within the triangle correspond to the time
with lower temperature than those are closer to the right side of the triangle (Fig. S9). This
could be explained by the increased biogenic OOA emissions when the temperature was higher,
as biogenic OOA tends to be distributed along the right side of the triangle (Canonaco et al.,
2015; Pfaffenberger et al., 2013). Also, when the temperature decreases, the increased biomass

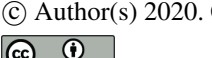



emissions make the OOA points to lie vertically within the triangle (Canonaco et al., 2015;
Heringa et al., 2011), which is the case for the winter data (Fig. 7b).
In July 2014, the rolling PMF LO-OOA moved towards the left side of the plot due to
increasing influences from $m/z$ 80, $m/z$ 94 ($C_2H_6S_2^+$), $m/z$ 95, and $m/z$ 96 (Fig. S7). Because the
OA signal of $m/z$ 80 is directly calculated from $m/z$ 94 (Allan et al., 2004), we did not
investigate the sources of $m/z$ 80. A potential source of these distinct ions in July is dimethyl
disulphide, which shows signals at $m/z$ 94, $m/z$ 95, and $m/z$ 96 (NIST Mass Spectrometry Data
Center, 2014). Dimethyl disulphide is widely used in pesticides. Considering that the sampling
site is in the middle of farmland, and the diurnal variation of $m/z$ 94 appeared to have peaks
during the daytime, we considered the LO-OOA in July to be highly affected by the agricultural
activities. However, the static factor profiles of summer LO-OOA from the seasonal summer
solution had much smaller intensities for $m/z$ 80 and $m/z$ 94 (Fig. S6), which enhanced the
scaled residuals for these two variables in the seasonal solutions.
In winter, LO-OOA (Fig. 9b) was highly affected by biomass burning emissions characterized
by the presence of $m/z$ 60, 73 (Alfarra et al., 2007), and the  LO-OOA position in the $f_{44}$ vs $f_{43}$
space moved towards the right top direction in the plot due to the increasing biogenic influence
as the temperature rose (Fig. 7b, Fig. S9) (Canonaco et al., 2015).
Figure 7 also highlights the advantages of rolling PMF over seasonal PMF due to its time-
dependent source profiles. For all the seasons, both seasonal and rolling results show that the
linear combinations of OOA factors could properly explain most of the measured OOA points.
However, with the static OOA factors for seasonal PMF solutions, it remains difficult to
capture the variabilities of some measured data points, while the rolling PMF OOA factors are
able to move correspondingly with the temporal changes of the clouds, which moves the factor
profiles closer to reality and potentially decreases the scaled residuals significantly (Fig. A3).





Figure S9 also shows the movements of LO-OOA and MO-OOA factor profiles monthly,
where LO-OOA moves towards the right direction as the temperature increases, except for the
two light blue squares (June and July) in Fig. S9a. It is clear that temperature plays an important
role for the positions of LO-OOA and MO-OOA in the $f_{44}$ vs $f_{43}$ space due to its influences on
the OOA sources (biogenic or anthropogenic) as well as the atmospheric processes, which is
consistent with previous studies in Zurich (Canonaco et al., 2015).

### 3.3.3 Statistical and rotational uncertainties

As suggested by Canonaco et al. (2020), combining the bootstrap resampling and the random
$a$-value techniques together with the rolling mechanism, we calculated the standard deviation
($\sigma$) and the mean ($\mu$) of the mass concentration for each data point from each OA factor in
selected "good" PMF runs. We estimated uncertainty of each OA factor using the slope of the
linear fit of $\sigma$ vs $\mu$. (Fig. 8.). Since the LOA factor was tightly constrained with an $a$-value of
0.05, it has the smallest variability (4%). Overall, we found relatively smaller errors of HOA,
BBOA, and MO-OOA (i.e., 18%, 14%, and 19%, respectively) and an error of 25% for LO-
OOA which is comparable with the previous study (Canonaco et al., 2020). The errors for both
the MO-OOA and the LO-OOA factor showed some temperature dependence. However, this
actually varied with time, and the errors did not significantly change when we separated the
dataset into four different temperature groups. Still, data points with higher temperature tend
to have larger error for the total OOA than with lower temperature (Fig. 8f). This is because
more complex aging processes for OOA factors at high temperature (>20 °C) can cause more
variability for the OOA factors.

### 3.3.4 Online vs. offline

The mass concentrations for HOA, BBOA, and total OOA were compared with corresponding
off-line AMS results (Vlachou et al., 2018) (Fig. S11). Despite some disagreement during
winter (BBOA and total OOA), BBOA showed a high correlation –with the offline results for
both $PM_{10}$ and $PM_{2.5}$, with $R^2$ of 0.83 and 0.84, respectively. The correlation for total OOA
was somehow lower, with $R^2$ of 0.31 and 0.46 for the offline results of $PM_{10}$ and $PM_{2.5}$ OOA,
respectively. The enhanced OOA concentration for the rolling results during winter season
compared to the offline SA results (Fig. 9a), as well as the differences between the rolling
results and the offline $PM_{2.5}/PM_{10}$ results regarding BBOA are most likely due to the fact that
the LO-OOA was heavily affected by biomass burning (Fig 9b). The offline results apportioned
this biomass burning affected LO-OOA into BBOA, whereas the online ACSM measurements
with a higher time resolution were capable to capture the fast oxidation process of biomass
burning sources. In addition, the rolling PMF technique enabled the LO-OOA factor profile to
adapt to the temporal viabilities of OA sources, so the relatively aged biomass burning related
sources was apportioned into LO-OOA during winter time by rolling PMF. The yellow line in
Fig. 9a depicts the mass concentration of $m/z$ 60 within LO-OOA, which clearly shows
significant enhancements during winter, as well as a good agreement with the LO-OOA time
series. HOA did not correlate at all, which may be expected because HOA is not water soluble,
and therefore has a very low recovery rate of 0.11 for the offline AMS technique based on the
previous study by Daellenbach et al. (2016).

## 4  Conclusions

In this study, we conducted the first rolling PMF analysis on a 13-month ACSM data collected
at a rural site of Switzerland. With the help of the a short rolling PMF time window together
with the random $a$-value and bootstrap resampling analysis, we obtained a time dependent SA
result with error estimations. Overall, we resolved a comprehensive 5-factor solution with
HOA, BBOA, LOA, MO-OOA, and LO-OOA. The contribution of HOA was constant during
the year (8.1–10.1%), while BBOA showed a clear seasonal variation (8.3–27.4%), which
peaked during winter (due to an increased residential heating source) and contributed least in



summer. OOA was a dominant source throughout the year with a contribution of 71.6% on a
yearly average. However, the biomass burning source had a strong influence on LO-OOA
formation in winter. Together with BBOA, they make residential heating a considerable source
at Magadino during winter. Therefore, a mitigation of residential wood combustion should be
considered for a reduction of PM levels in Magadino, especially in winter.
This manuscript also provided a recommended criterion list (Table S1) as well as a novel way
to define thresholds with minimum subjective judgements (student's $t$-test), which could be a
leading example for other SoFi Pro users to conduct rolling PMF. To ensure a good
representation of the modelled POA factors and to validate the SA results, we also used the
correlations between the PMF factor time series and external data. Both HOA and BBOA
agreed well with the corresponding external tracers ($NO_x$, $eBC_{tr}$, and $eBC_{wb}$) for the yearly
cycles, except summer. This is because the aethalometer model for eBC SA has higher
uncertainties with smaller $eBC_{wb}$ mass concentrations. Also, $NO_x$ could originate from multiple
sources in this season. Therefore, we used HOA vs. eBC and $EV_{60,BBOA}$ to justify these two
factors in summer. The correlation of HOA vs eBC had an $R^2$ of 0.28, with an $EV_{60,BBOA}$ of
0.55 in summer. Moreover, the MO-OOA and LO-OOA factors correlated well with inorganic
$SO_4$ and $NO_3$, respectively. The identified primary and secondary OA factor profiles were
consistent with the OA factors previously found at a variety of urban, rural, and remote
European locations.
This paper assessed the statistical and rotational uncertainties of the PMF solution by
combining the bootstrap resampling technique and the random $a$-value approach. It shows
relatively small errors for constrained factors compared with a previous study in Zurich
(Canonaco et al., 2020), and comparable errors for the OOA factors.



We also presented a head-to-head comparison between seasonal PMF solutions and the rolling
PMF solution. The POA factors showed good agreement between seasonal and rolling PMF
solution, while the OOA factors exhibited greater differences. Overall, the rolling PMF
retrieved a somewhat better solution in terms of agreement with external tracers, but much
better correlations between the OOA factors and corresponding inorganic salts. In addition, the
rolling PMF results provided more realistic results by adapting the temporal variations of OOA
factors in the $f_{44}$ vs $f_{43}$ space, which also led to much smaller scaled residuals than for the
seasonal PMF. The time series of BBOA and total OOA agreed well with those from offline
AMS AS results (Vlachou et al., 2018), except for winter when the fast oxidation processes of
biomass burning emissions were not captured by the offline AMS technique.
Knowledge of diurnal, seasonal and annual changes in OA sources is essential for interpreting
the yearly cycles of OA and defining mitigation strategies for air quality. With the help of more
accurate and realistic OA sources together with an estimation of the statistical uncertainty of
PMF more constraints can be provided both for climate and air quality models. These improved
results are therefore highly valuable for policy makers to solve aerosol-related environmental
issues.
# 5   Appendix A: Comparison between seasonal and rolling PMF
solutions
The bootstrapped seasonal PMF solutions were compared with the full year rolling PMF results
as follows. For each factor, the correlations with external data, the ion intensities in the factor
profiles, and the mass concentrations retrieved from the two different source apportionment
techniques were compared. The correlations of the factor time series with external data (i.e.,
NO$_x$, eBC$_{tr}$, eBC$_{wb}$, eBC$_{totoal}$, SO$_4$, NO$_3$, and NH$_4$) are presented in **Table A1**. The rolling
results showed generally slightly better correlations between LO-OOA and NO$_3$, MO-OOA





and $SO_4$, and total OOA with $NH_4$ than the seasonal PMF results, which is consistent with the
comparison results from Canonaco et al. (2020). A significant improvement was evident for
LO-OOA vs $NO_3$ in spring (with $R^2$ increasing from 0.02 to 0.48). Concerning the correlations
of POA factors with external data, rolling results and seasonal showed similar results
**Table A1** Correlation coefficients ($R^2_{pearson}$) between the factor contribution and expected
tracers over the year and for individual meteorological seasons (p<0.05).

| Factor | Yearly | | SON_2013 | | DJF | | MAM | | JJA | | SON_2014 | |
|---|---|---|---|---|---|---|---|---|---|---|---|---|
| | Seasonal | Rolling | Seasonal | Rolling | Seasonal | Rolling | Seasonal | Rolling | Seasonal | Rolling | Seasonal | Rolling |
| HOA / $NO_x$ | 0.37 | 0.35 | 0.52 | 0.5 | 0.46 | 0.47 | 0.34 | 0.36 | 0.15 | 0.15 | 0.44 | 0.42 |
| HOA / $eBC_{tr}$ | 0.34 | 0.33 | 0.29 | 0.35 | 0.41 | 0.42 | 0.39 | 0.31 | N/A | N/A | 0.38 | 0.39 |
| HOA / eBC | 0.55 | 0.51 | 0.79 | 0.77 | 0.77 | 0.73 | 0.5 | 0.41 | 0.29 | 0.28 | 0.5 | 0.47 |
| BBOA / $eBC_{wb}$ | 0.82 | 0.82 | 0.81 | 0.79 | 0.84 | 0.81 | 0.67 | 0.6 | N/A | N/A | 0.3 | 0.27 |
| MO-OOA / $SO_4^{2-}$ | 0.58 | 0.49 | 0.49 | 0.61 | 0.52 | 0.49 | 0.62 | 0.66 | 0.63 | 0.57 | 0.43 | 0.46 |
| LO-OOA / $NO_3^-$ | 0.11 | 0.32 | 0.28 | 0.42 | 0.28 | 0.23 | 0.02 | 0.48 | 0.33 | 0.36 | 0.19 | 0.29 |
| OOA/ $NH_4^+$ | 0.46 | 0.44 | 0.52 | 0.55 | 0.34 | 0.26 | 0.73 | 0.75 | 0.48 | 0.47 | 0.57 | 0.59 |


As shown in **Fig. A1** Comparison of the mass concentrations resulting from rolling PMF
and from the seasonal analysis for each factor (colour coded by date and time)., which
shows a good agreement for two techniques, except for MO-OOA and LO-OOA. In general,
the slope of 1.09 for rolling total OOA vs seasonal OOA suggests a slight underestimation of
the OOA contribution by the seasonal PMF solutions, while the slope (<1) for HOA and BBOA
suggests that the seasonal PMF solutions overestimate HOA and BBOA. In addition, LOA
shows the best agreement between the seasonal and rolling solutions, due to the tight constraint
of LOA with an *a*-value of 0.05.
The LO-OOA and MO-OOA factors showed worse agreement than the POA factors for the
whole dataset. They had good correlations in each meteorological season, however, with
different slopes. For instance, seasonal PMF underestimated LO-OOA in spring and fall 2014,
but both seasons showed high correlation with rather narrow scattering. The underestimation
of LO-OOA by seasonal PMF was compensated by the overestimation of MO-OOA for these



two seasons, therefore, the summed OOA still showed a high correlation between rolling and
seasonal PMF results. This is expected, as the rolling PMF allows the source profiles to adapt
to temporal variations, while seasonal PMF only has static source profiles.

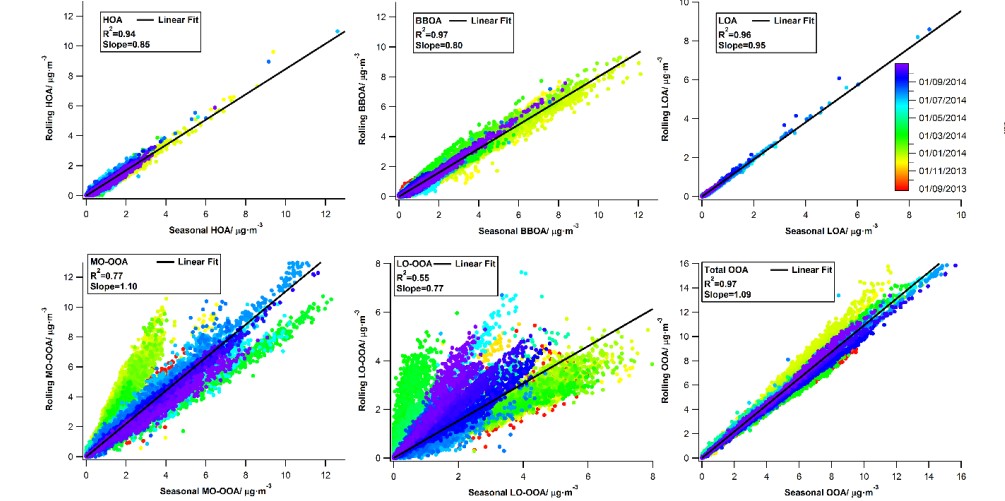


**Fig. A1** Comparison of the mass concentrations resulting from rolling PMF and from the
seasonal analysis for each factor (colour coded by date and time).

The differences in the major variables of the OOA factors (i.e., *m/z* 44, 43, and 60) shifted the
mass concentrations significantly. Therefore, we also compared the factor profiles for both
techniques (**Fig. A2**). For instance, LO-OOA during spring showed higher intensity at *m/z* 44
for the rolling PMF results than for the seasonal PMF results (**Fig. A2**), which caused the
underestimation of LO-OOA for the seasonal PMF in spring. When we averaged the total OOA
factor using mass-weighted MO-OOA and LO-OOA factors, rolling PMF yielded higher *m/z*
60 for all seasons. As a result, seasonal PMF slightly underestimated the summed OOA factors
by around 9%, but slightly overestimated the POA factors by less than <6%.



The profiles of the constrained factors (HOA, BBOA, LOA) from the rolling results show very
high correlation with the seasonal results (**Fig. A2**), which suggests that the primary factors
and the tightly constrained factor (LOA) were consistent with the static profiles from the
seasonal PMF analysis.

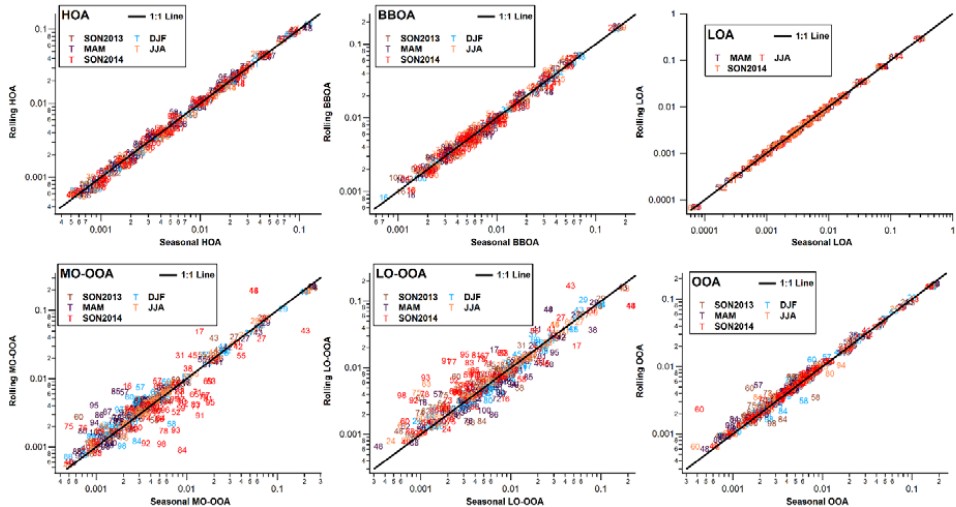


**Fig. A2** Profile comparisons between rolling results and seasonal results for each factor (log
scale).

We compared the scaled residuals from both source apportionment techniques (**Fig. A3**). The
rolling PMF solution had smaller scaled residuals (narrower histogram and the centre was
closer to 0) than that of the seasonal PMF solution, which is expected because rolling PMF had
more flexibility to adapt to the temporal variabilities of the OA sources.




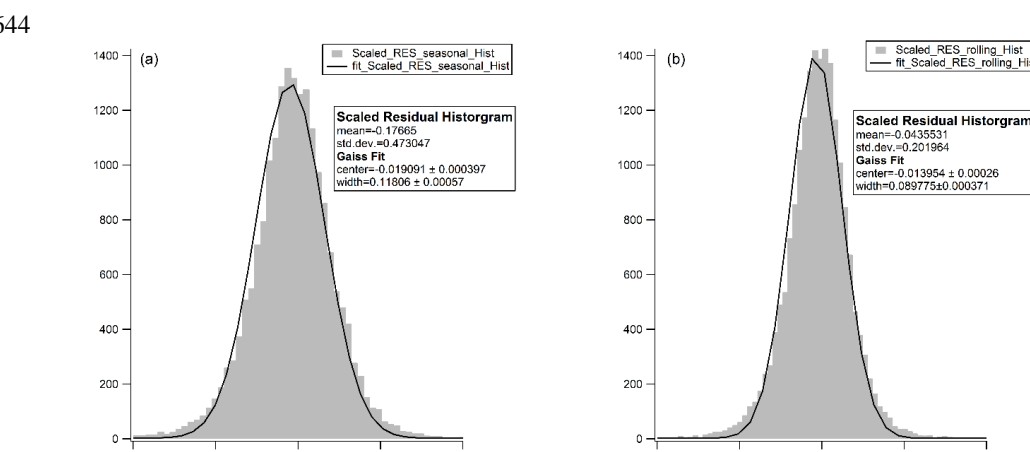


**Fig. A3** Distribution of the scaled residuals over the whole year for the seasonal solution (a) and rolling solution (b).

Summarizing, HOA and BBOA were consistent for both rolling and seasonal PMF analysis in terms of the time series, correlations with external tracers, and factor profiles due to the consistency of their chemical factor profiles. In contrast, the MO-OOA and LO-OOA factors were more scattered in terms of averaged factor profiles and mass concentration, which suggests that seasonal PMF analysis was not sufficient to capture these temporal variabilities of their oxidation processes. Also, rolling PMF showed smaller scaled residuals. Therefore, we conclude that the rolling PMF analysis provides more realistic results than the seasonal analysis.

## Data Availability

The data are available upon request to the corresponding author.

## Competing interests

The authors declare no competing interests in any form for this work.



## Author contributions


G. C. and Y. S. contributed equally for this manuscript. G. C. wrote the manuscript, illustrations
as well as data treatments and processing. Y.S. wrote the preliminary manuscript and analysed
preliminary results. R. F. and P. G. helped to run the campaign. P. G., and C. H. provided
external data to validate PMF solution. F.C. provided technique support for SoFi Pro. F.C., A.
T., K. R. D., A. V., J.G.S., I. EI. H., U. B., and A. S. H. P. participated discussions for this
study.

## Acknowledgements


The ACSM measurements were supported by the Swiss Federal Office for the Environment
(FOEN). The leading role of the Environmental group of the Swiss Federal Laboratories for
Materials and Testing (Empa) in supporting the measurements is very much appreciated. Y. S.
acknowledges supports by the "Wiedereinsteigerinnen Program" at the Paul Scherrer Institute.
This study was also supported by the cost action of Chemical On-Line cOmpoSition and Source
Apportionment of fine aerosol (COLOSSAL, CA16109), a COST related project of the Swiss
National Science Foundation, Source apportionment using long-term Aerosol Mass
Spectrometry and Aethalometer Measurements (SAMSAM, IZCOZ0_177063), as well as the
EU Horizon 2020 Framework Programme via the ERA-PLANET project SMURBS (grant
agreement no. 689443).

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

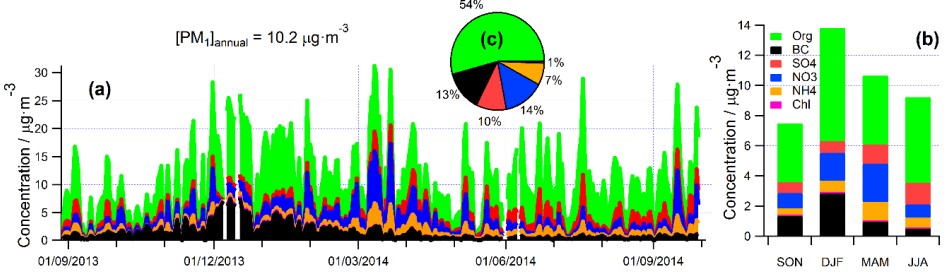


**Fig. 1** Chemical composition of PM$_1$ in Magadino 2013-2014 – daily (a), seasonal (b) and

annual (c) averages. The labels indicate the non-refractory organics (Org), sulphate (SO$_4$),

nitrate (NO$_3$), ammonium (NH$_4$) and chloride (Cl) ions measured by ACSM, and the black

carbon (BC) measured by light absorption.

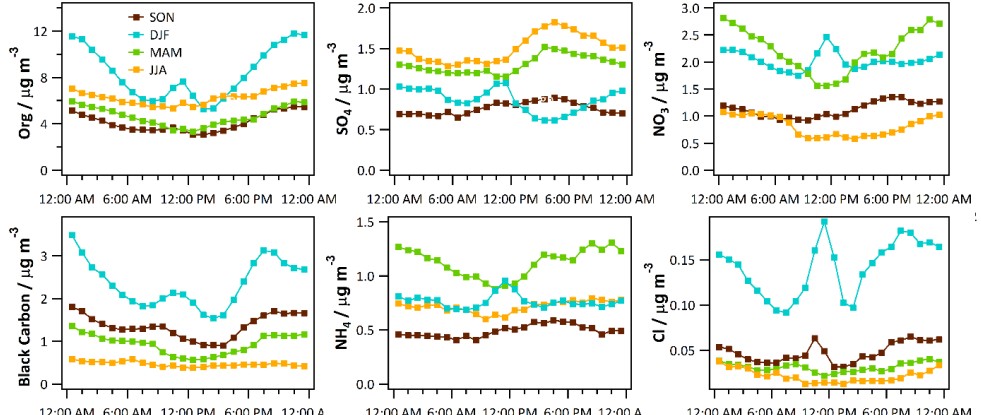

990

**Fig. 2** Seasonal diurnal cycles of PM$_1$ constituents calculated as an hourly average for ACSM

organic and inorganic species (sulphate, nitrate, ammonium, and chloride) and black carbon





**Fig. 3** Annual cycle of OA sources: (a) absolute and (b) relative OA contributions plotted as

30-min resolved time series, (c) BC source apportionment.



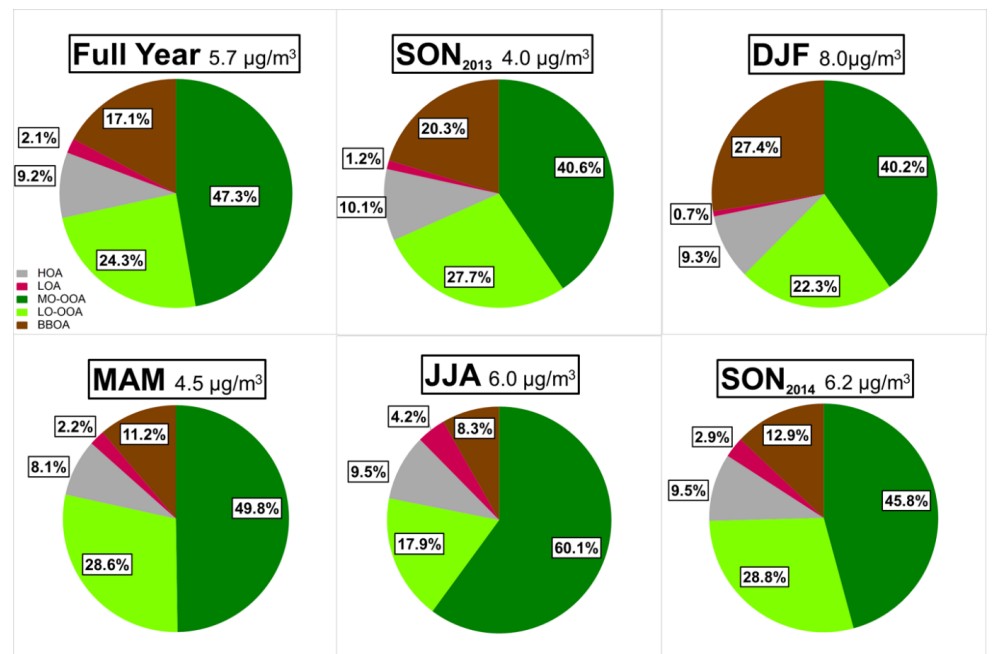

**Fig. 4** OA pie charts for the whole year and for different seasons.



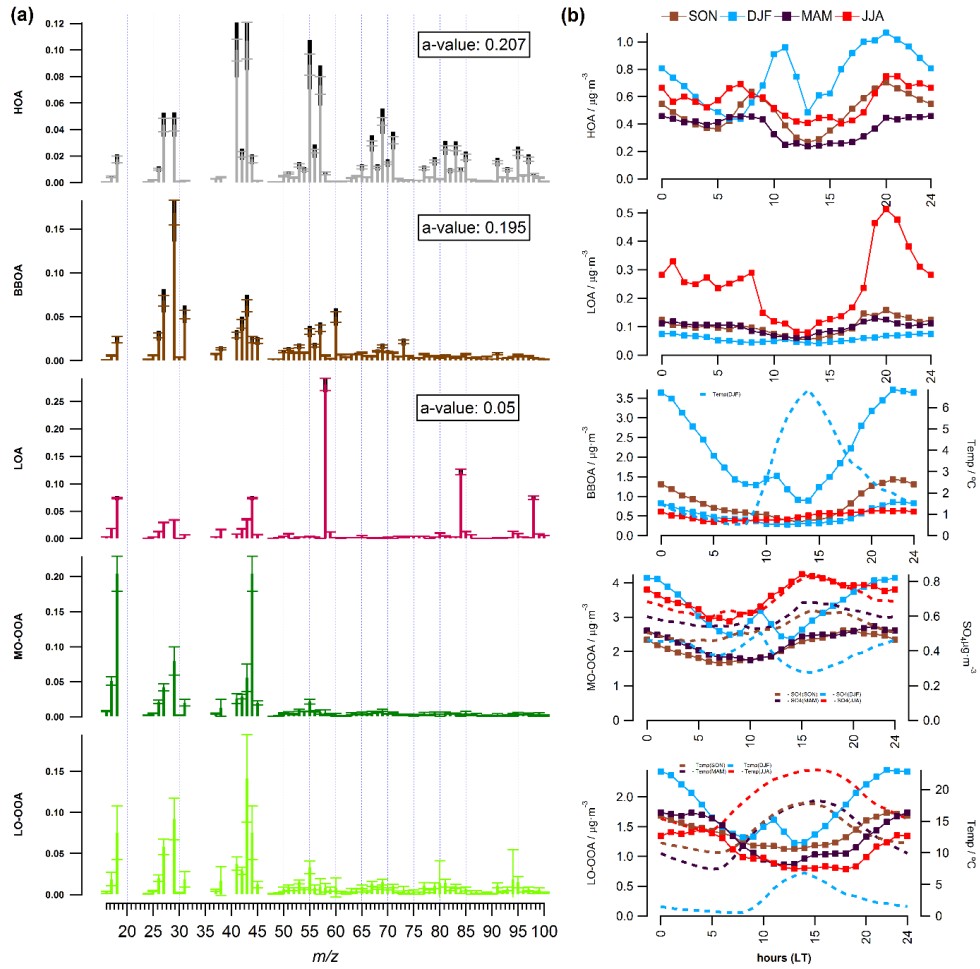

998

**Fig. 5** Overview of the primary and secondary OA sources in Magadino in 2013-2014: (a) OA factor profiles and (b) seasonal diurnal cycles of HOA, BBOA, LOA, MO-OOA, and LO-OOA. The ambient temperature is shown on the LO-OOA diurnal plots, respectively. In (a) the error bar is the standard deviation; the black bars show the maximum and the minimum that the variable allowed to be vary from the reference profiles. The average, 10th and 90th percentiles for a-values of HOA are, 0.195, 0.007 and 0.378, respectively. Also, the average, 10th and 90th percentiles for a-values of BBOA are 0.202, 0.025 and 0.379, respectively.

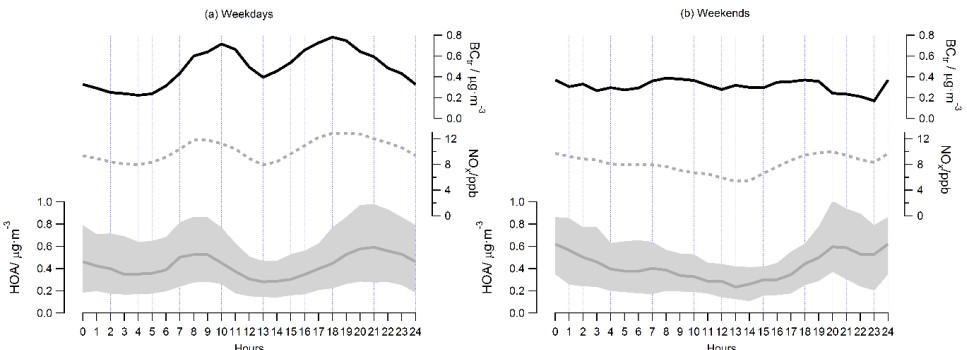


**Fig. 6** Diurnal cycles of HOA (grey symbols), black carbon apportioned to traffic emissions

eBC$_{tr}$ (dashed lines) and NO$_x$ (dotted lines) for weekdays (a) and weekends (b). The shaded

areas represent interquartile range for 1-hour average HOA.

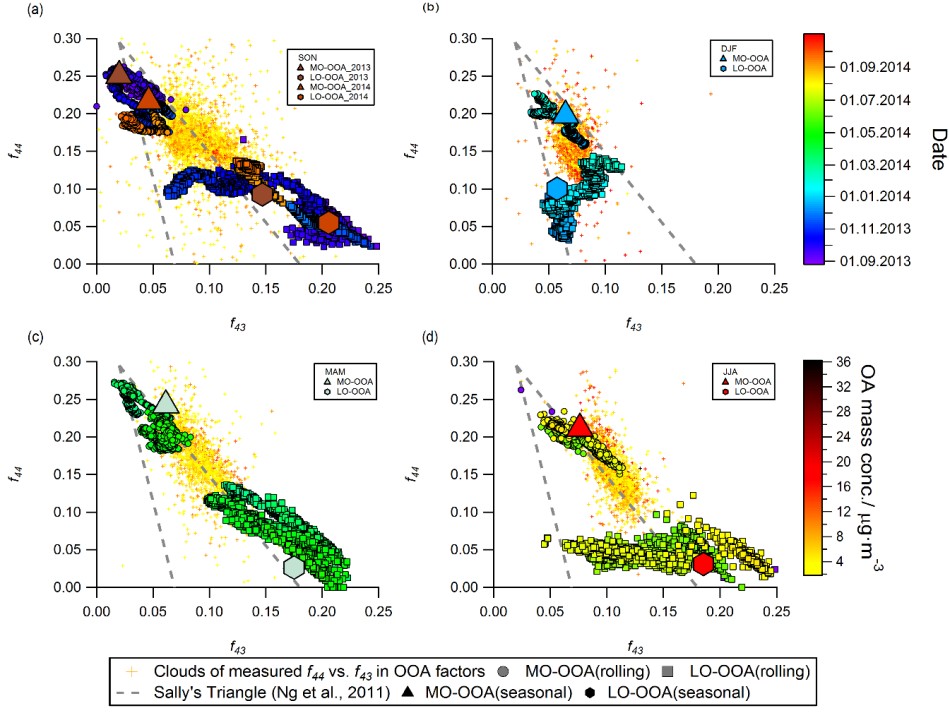


**Fig. 7** OOA $f_{44}$ and $f_{43}$ for four different seasons. The yellow cloud of data points represents the

$f_{44}$ vs $f_{43}$ by subtracting the $f_{44}$ and $f_{43}$ contributed from HOA, BBOA and LOA factors. They

are color coded by the total OA mass concentration. The circles, triangles, and squares





represent the ratio between $f_{44}$ and $f_{43}$ intensities within the factor profiles of MO-OOA and LO-
OOA, respectively. While the smaller size of circles, triangles, and squares are from rolling
PMF analysis, which are color coded by the date and time. The dash line are the Sally's triangle
from (Ng et al., 2011) and depicts the region where several PMF OOA from the last decade
resided in the $f_{44}$ vs $f_{43}$ space.






**Fig. 8** Absolute statistical uncertainties of PMF for HOA, BBOA, LOA, LO-OOA, MO-OOA
and total OOA (LO-OOA+MO-OOA) for all data. The data points colour-coded all data points



by temperature. The PMF error (uncertainties) of selected PMF runs and rotational
uncertainties is estimated using the slope of the linear regression of standard deviation (σ) vs.
the averaged mass concentration (μ) for each factor.

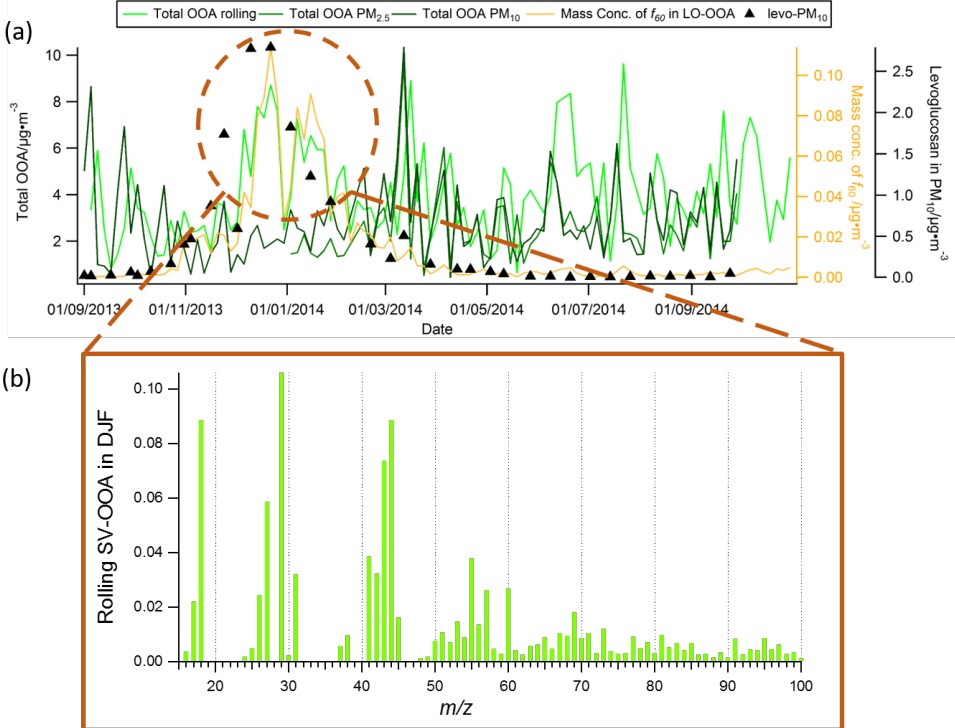


**Fig. 9** (a) Time series of total oxygenated organic aerosol (LO-OOA+MO-OOA) from online
and offline source apportionment solutions, together with f60 in LO-OOA for online solution,
and levoglucosan in PM10 filter; (b) Averaged LO-OOA factor profile from online solution
during DJF (Dec, Jan, and Feb), when online total OOA is significantly higher than that of
offline solutions.