# Peer review of "Time dependent source apportionment of submicron organic"

_Atmospheric Chemistry and Physics, 2020_

## Referee Comment (RC1) · Anonymous Referee #1 · 2 Feb 2021

The manuscript by Chen et al. describes the Organic Aerosol (OA) Source Apportionment results of one year long measurements carried out by an aerosol chemical speciation monitor (ACSM) in Magadino, a rural village located in the south of the Swiss Alpine region, one of the most polluted areas in Switzerland. The Authors applied positive matrix factorization (PMF) and identified well known OA factors: two primary OA factors (traffic-related hydrocarbon-like OA, HOA and biomass burning OA, BBOA), and two secondary factors (a less oxidized oxygenated OA, LO-OOA factor, and a more oxidized oxygenated OA, MO-OOA factor) plus a socalled local OA (LOA) factor. The main novelty of the study is the application (for the first time on a rural site) of a new "rolling" algorithm to account for the temporal changes of the source profiles in PMF.

Thus, in addition to the description of the main components that characterize OA in the site under examination, the manuscript presents a detailed comparison between the more traditional methods of applying PMF and the new "rolling" method suggested as the best approach to improve the representativeness of the factors identified. The Authors also calculated the uncertainties ( $\sigma$ ) for the modelled OA factors (i.e., rotational uncertainty and statistical variability of the sources) finding values ranging from a minimum of  $\pm 4\hat{a}\dot{A}\dot{L}$ % (LOA) to a maximum of  $\pm 40\hat{a}\dot{A}\dot{L}$ % (LO-OOA). The manuscript represents a huge effort with the aim of improving OA source apportionment PMF capability and it is of possible interest for a large audience of the atmospheric organic aerosol community. However, the text results guite hard to read and it is not completely clear in some parts because of a low-quality presentation and organization of figures and sections (both in the main text and in the SI). Moreover, the significance of the improvement provided by the new method should be clarified better (given the huge amount of analyses required to apply it). For this reason, my suggestion is to accept the paper only after a strong re-organization of the text and after the consideration of some major issues listed below.

Major issues and general comments:

The amount of data presented, the number of different methodologies and settings together with their evaluations make already complex to follow the discussion and understand the main results. Additionally the text is fragmentary, with too many mixing up with SI (sometimes with wrong references). All this makes too hard to follow the discussion and to find out the scientific relevance of the whole work. I strongly recommend to re-organize the paper, summarizing in the main text the most fundamental approaches and results and leaving technicalities to the SI (which should be better organized as well).

The rolling analysis is for sure an interesting attempt to deal with the temporal changes of the source profiles in PMF running over long-term datasets. On the other hand, it requires a huge amount of PMF runs (several thousands) which are very time con-
suming and expensive in term of calculation resources. This can be particularly true if someone hypothesize to extend the same approach to high-resolution AMS data or other datasets in which the number of variables can be substantially higher. Moreover, it is not clearly quantified the improvement of using the rolling approach with respect to the seasonal: I mean, if the the rolling is leading to estimations with uncertainties up to 40% for OOA factors and the improvement in quantification of the same factors is of the order of 5-10% someone can argue that the game is not worth of the effort. I'm not saying this is true, but in my opinion the Authors should add a more comprehensive (even if synthetic) description (motivated by numbers) of the advantages/disadvantages of the rolling approach together with recommendations on how and when the approach can be profitably used or not.

Specific comments

Abstract P2, L22: "cite" probably misspelled for "city"

P2, L27-28: the sentence is redundant: it was already introduced few lines above the distinction of OOAs in two main types.

Introduction P4, L83: here as well as along all the text, please revise the references: you should re-order them based on the years of publication.

P7, L144-146: where is the comparison? Reading this sentence, it seems in Fig. S1 we should find the comparison between different CE, but it's not the case. Please rephrase the sentence clarifying better what is presented in Fig. S1. Consider also that Fig. S1b is not introduced at all in the text neither explained in SI.

P11, L230 and all Section 2.6: there is a lot of confusion between this section and the Supplementary Section 2. It is difficult to follow and understand the steps and what is happening. For example, you mention here the socalled Local OA factor (LOA) but you introduce it much later in the text (and, since it is not a standard factor, it is difficult to understand the motivation of the described procedure). Moreover, you
discuss figures of SI starting from Fig S6 here, and only later the previous ones. For this reason, please consider to reorganize substantially the text between this section and the corresponding SI section.

P16, L356: again, LOA factor is discussed but it is not introduced/described yet. Please introduce it before commenting on it: since it is not a standard factor you would need to add spectra and explanations before.

P17, L359-364: which kind of tests? It is not clear what you have done here. Consider to rephrase or to remove. Moreover, are you sure you are citing the right figure here? Fig S6 looks unsuitable to me here.

P17-18, L380-391: Consider to anticipate this introduction of LOA factor, as mentioned before. Moreover, why do you call this factor as "Local OA"? If I understood well it is clearly the result of an instrumental artifact: it is interesting that PMF can isolate also this kind of problems, but you should explain and put the emphasis on this.

P18, L394: Supplementary Section 4 is introduced and discussed before Section 3. The Supplementary Figure (Fig. S10) discussed here is instead postponed. This is misleading and create confusion.

P19, L422-424: why do you introduce Fig. 7 before Fig. 6? This sentence is anticipating the topic creating confusion. Please rephrase or remove it.

P20, L436-437: Could you please elaborate more on the peaking concentrations at 10-11am? Why the daily maximum is so late in the morning? I'm used to thinking that the diurnal evolution of PBL leads to more dilution and so lower concentrations of pollutants and that it starts earlier in the morning. I suppose there is a late sunlight illumination of the ground at the site, but it should be explained clearly because it sounds weird.

P23, L519-521: what do you mean with "more complex aging processes"? It is not clear and/or highly speculative. It is actually demonstrated by an increasing number of studies that OOA formation and ageing is complex also under low temperature and
dark conditions. If you mean that the higher variability at higher temperature could reflect different sources/precursors emissions enhanced at higher temperature (e.g., higher emissions of biogenic VOC and the subsequent mixing with preexisting OOAs or whatever) you should explain it better.

P24, L528-534: this is interesting. But the sentence is quite problematic. Can you elaborate more on it? Do you believe it can be also a question of selective water solubility of components?

Conclusions: expressions like "a somewhat better solution" or "more realistic results" are quite subjective and vague. "More realistic" based on what? What is the real improvement of using rolling PMF instead of more traditional (and less time-consuming) methodologies in term of identification and especially quantification of OA sources? You should clarify it better here in the conclusions. And possibly you should add a more comprehensive description (motivated as much as possible by numbers) of the advantages/disadvantages of the rolling approach together with recommendations on how and when the approach can be profitably used or not.

Figures General low quality, with too small font sizes making difficult to read labels of the axes and legends. Sometimes problematic also the choice of colors (e.g., Fig. 5 and 9a). Please check the readability of all the figures in the main text, in the Appendix A and in SI.

Figure 3: given that the x-axis (the time period) is common to all the panels, please consider to use only one or at least to make them of the same length (in order to improve the readability and the comparison between the different time trends)

Figure 5: all the labels (axes name and values, legend, etc.) are difficult to read. Especially graphs in panel b are completely not aligned, their legend is unreadable and the colors of the time series are misleading. Please increase the font size of all legends and labels and improve the general format of the figure.

ACPD
Supporting Information Check if the Figures follow the order of presentation in the text: Figure S6 should be anticipated (because discussed before, at the beginning of Section S2). In order to improve the readability it is also important to put intervals and possibly titles between subsections or figures/tables referring to different tests/results/data.

Fig. S1b is not introduced at all in the main text neither explained here in SI. You need to explain what it is showing (for instance, what is the mini-denuder? Where it is introduced?)

Section 2 P5, L62-63: do you have any references for this? What do you mean with "more accurate estimations"?

P6, L88-91: it is hard to understand how do you use t-test. This is probably better explained in Canonaco et al. 2020, but it is important to spend some more words also here to improve the understanding of the readers.

P6, L96-99: The same comment, it is hard to understand the procedure. Please rephrase and explain better.

Fig. S5: labels of the axes are missing. Please add them and increase the font size of the color-legend.

---

## Referee Comment (RC2) · Anonymous Referee #2 · 6 Mar 2021

Review comments, This manuscript reports the analysis of one year of ACSM mass spectral data obtained from a polluted rural Swiss site using ME-2 implemented in the time-rolling scheme. Five factors were resolved, including two POA factors (traffic-related HOA and BBOA), one local OA (LOA) factor, and two oxygenated OA factors (a less oxidized LO-OOA and a more oxidized MO-OOA). The diurnal cycles, seasonal variations, sources and processes of the OA factors were discussed and the statistical and rotational uncertainties for the modelled OA factors were assessed. In addition, the rolling PMF analysis results were compared to the results from the conventional PMF analysis on the data segregated by seasons and the source apportionment of offline AMS filter samples. This is a valuable study that demonstrates the utility and

strength of applying rolling PMF analysis to the long terms ACSM data. This paper is an important contribution to the field of aerosol source apportionment and should be accepted for publication after the following review comments are addressed.

This study is extensive and the amount of information given in the manuscript is massive. However, the texts can sometimes be a bit hard to follow and confusing. I suggest more efforts are made to organize the contents more effectively, streamline the discussions, and improve the paper's general readability. For example, the descriptions of various aspects of the PMF analysis are lengthy and somewhat lack of coherence. Compiling a summary table of the key information and parameters used in the analysis could be useful. Many figures in the manuscript and the supporting information are hard to read and the figure captions are ineffective. Font size should be increased to be legible under normal page view. Figure captions should be sufficiently detailed to make the figures understandable.

The LOA factor appears to be an artifact arising under certain instrumental condition, thus is not a real ambient OA factor per se. Calling it Local OA implies that it is an OA component associated with certain local emission. This could be misleading, so are the discussions about the ambient behaviors of this factor.

The effort to determine statistical and rotational uncertainties for the OA factors is commendable. One question however is since the PMF solutions are selected, the average a-value is calculated to be fairly low (around 0.2). The errors for the OA factors are then determined based on the selected PMF solution. Aren't this approach somehow circular?

Specific comments:

Line 22, Change "cite" to "site".

Line 21 - 22, it is not appropriate to claim this study "the first ever application of rolling PMF analysis for a rural site . . ." The data analyzed by Parworth et al. (2015) came

from a rural site in Central United States and that study was the first to report the application of rolling window PMF analysis on long term ACSM data.

Section 2.2, it is useful to provide information about ACSM operation and quality control measures, such as the ACSM measurement time resolution, the detection limits for NR species.

Line 148, change "doesn't" to "don't"

Line 179, what's the reason for the ion signals at m/z 12 and 13 being mostly negative? Is this an issue specific to this study?

Line 179 -180, why is m/z 15 "affected by high biases due to potential interference with air signals"? ACSM determines particle signals as the difference between the filter-off and filter-on modes. So, aren't the air influences on the ACSM diff signals removed from the diff signals?

Line 251, the meaning of "within a range of 0.4" is vague. Spell out the range. What exactly does "random" mean in "a random a-value"?

Section "2.6.1 Window settings", many discussions within this section do not seem closely related to the topic of setting the proper rolling window size.

Line 254- 255, it is mentioned that different a-values were chosen to constrain the fitting of LOA. is there any significance with the specific value?

Line 257, does this sentence suggest surface ionization enhances the production of N-containing ions? Are there reference(s) to support this?

As the LOA factor appeared after a filament change, it is a factor associated with a certain instrument condition. ME-2 analysis that constraints the LOA profile may lead to the retrieval of this factor may be forced even when it was not supposed to be present.

Line 277, what is eBCtr ?

Line 282, isn't the measurements done with Q-AMS, how was it known that m/z 43 is C2H3O+?

Line 287 – 288, what do 4th and 5th position reference to?

Line 316, define BLH

Line 325-328, this sentence is awkward and difficult to understand. Consider to revise.

Line 398, the small seasonal differences in HOA% is interesting. What's the explanation? Since HOA is mainly a POA factor whose concentration should be influenced strongly by BLH, it loading tends to be much higher during winter than in summer. In contrast, stronger photochemistry tends for lead to higher SOA in the summer. So the seasonal difference in HOA% is expected to be strong in winter.

Line 428, what are the standard deviations for the a values?

Line 483, is dimethyl disulphide sufficiently low volatility to be in the particle phase? Are there HR-AMS or other analytical results to support the presence of this compound? As pesticide application is usually seasonal, did you see evidence from this perspective?

Line 519 – 521, are there references to support this statement?

Fig 3, what do BCtr, BCwb stand for?

Fig. S6, explain the error bars in the caption

Fig S8, how exact was the probabilities calculated?

Fig. S9, the key is difficult to understand, what are the dots exactly, what does it mean "clouds of measured f44 vs. f43 in SOA factors"?

Fig. S10, figure caption hard to understand. What are the "Missing time points"?
* * *

---

## Author Comment (AC1) · 19 Jul 2021

We thank all the constructive comments from two reviewers. The following texts are the response to the reviewers.

The normal italic font is original *reviewer comments,* smaller green font is authors' responses, and the small blue italic font is *the changes in the revised version.*

**Anonymous Referee #2**

*Review comments, This manuscript reports the analysis of one year of ACSM mass spectral data obtained from a polluted rural Swiss site using ME-2 implemented in the time-rolling scheme. Five factors were resolved, including two POA factors (traffic-related HOA and BBOA), one local OA (LOA) factor, and two oxygenated OA factors (a less oxidised LO-OOA and a more oxidised MO-OOA). The diurnal cycles, seasonal variations, sources and processes of the OA factors were discussed and the statistical and rotational uncertainties for the modelled OA factors were assessed. In addition, the rolling PMF analysis results were compared to the results from the conventional PMF analysis on the data segregated by seasons and the source apportionment of offline AMS filter samples. This is a valuable study that demonstrates the utility and strength of applying rolling PMF analysis to the long terms ACSM data. This paper is an important contribution to the field of aerosol source apportionment and should be accepted for publication after the following review comments are addressed.*

*This study is extensive and the amount of information given in the manuscript is massive. However, the texts can sometimes be a bit hard to follow and confusing. I suggest more efforts are made to organise the contents more effectively, streamline the discussions, and improve the paper's general readability. For example, the descriptions of various aspects of the PMF analysis are lengthy and somewhat lack of coherence. Compiling a summary table of the key information and parameters used in the analysis could be useful. Many figures in the manuscript and the supporting information are hard to read and the figure captions are ineffective. Font size should be increased to be legible under normal page view. Figure captions should be sufficiently detailed to make the figures understandable.*

We thank the reviewer for the positive feedback. We also think this study will be very important to the community and will be a good role model for similar analyses.
Our apology for the poor presentation quality, therefore, we have put in an extensive effort to reorganise the structure of the manuscript in the revised version for both the main text and the SI. Specifically, we moved most of the methodologies and settings into the SI as suggested, and we have added a table of content for the SI to provide a good overview of the paper. Also, sequences of the figures in the SI were reordered to follow the storyline for both the main text and the SI. All the figures were reconstructed in the revised version based on the reviewer's specific comments.

*The LOA factor appears to be an artifact arising under certain instrumental condition, thus is not a real ambient OA factor per se. Calling it Local OA implies that it is an OA component associated with certain local emission. This could be misleading, so are the discussions about the ambient behaviors of this factor.*

We agree that LOA could lead to a misunderstanding of its source, therefore, we took the reviewer's suggestion and changed the name from LOA to *m/z* 58 related OA (58-OA) factor. The surface ionization changes due to the switch of the filament in the middle of the campaign and constraining of this factor during PMF analysis led to an overestimation of its mass concentration. The time series of it should be always considered as the upper limit of this source contribution, and the real mass concentration of it could be substantially lower. However, since the mass concentration remains low, we still consider it as a minor factor. Therefore, we only include it in the PMF analysis without further exploration of its potential source. To better explain the 58-OA factor, we therefore modified the introduction of this factor in Section 2.5 (revised version) as follows: "*The 58-OA was dominated by nitrogen-containing fragments (at m/z 58, 84, and 98). In general, ACSM estimates organic m/z 98 signal by dividing organic m/z 84 to a factor of 2 according to the*

*fragmentation table of organic species that was provided by Allan et al. (2004). Thus, the intensity of m/z 98 is always half of the intensity of m/z 84 in each factor. This 58-OA only appeared after the filament was switched on 14 April 2014. The instrument setup thus strongly influenced the sensitivity of these components (likely due to influences of surface ionizsation). The nitrogen-containing ion, m/z 58, was also observed in Hildebrandt et al. (2011) due to the enhanced surface ionisation in a certain period. In addition, the potassium signal enhanced at the same time, which further corroborated our hypothesis of the enhanced surface ionisation. Also, since this factor was constrained through the whole dataset, the PMF model overestimated the mass concentration of this factor significantly, which leads to high uncertainties for the 58-OA. Therefore, the time series of this source should be considered as the upper limit, and the real mass concentration of it could be substantially lower. However, with the low mass concentration of the 58-OA during the whole campaign, we considered it as a minor factor. Thus, this factor was considered in the PMF analysis, but no further interpretation of its potential source will be attempted in this manuscript.*"

*The effort to determine statistical and rotational uncertainties for the OA factors is commendable. One question however is since the PMF solutions are selected, the average a-value is calculated to be fairly low (around 0.2). The errors for the OA factors are then determined based on the selected PMF solution. Aren't this approach somehow circular?*

The averaged employed *a*-value is around 0.2, but the upper *a*-value we allowed the model to vary is 0.4. Also, the employed *a*-value only give us a sense of the variabilities of OA source profiles. It does not necessarily mean the higher the averaged *a*-value, the higher the errors are. In addition, it is always worthwhile to check the distribution of employed *a*-values of these constrained factors. Because it can help us to cross-validate if the constrained factors have rather small errors, for example, when the error of BBOA is rather big, and there are quite many selected runs distributed at the highest a-value range, it could suggest that the upper *a*-value applied during bootstrap and rolling analysis is not sufficient. More freedom is required to obtain a better resolved BBOA factor. Moreover, this study combined the bootstrap re-sampling and rolling technique, therefore, it is also very important to understand the uncertainty in a more quantitative way using the time series of OA factors. Thus, we think both approaches are essential, and we would like to keep them both.

*Line 22, Change "cite" to "site".*

Corrected

*Line 21 - 22, it is not appropriate to claim this study "the first ever application of rolling PMF analysis for a rural site . . ." The data analysed by Parworth et al. (2015) came from a rural site in Central United States and that study was the first to report the application of rolling window PMF analysis on long term ACSM data.*

We appreciated the comment from the reviewer. We rephrased the sentence as follows: "*As the first-ever application of rolling PMF with ME-2 analysis on a yearlong dataset that is collected from a rural site,….*"

*Section 2.2, it is useful to provide information about ACSM operation and quality control measures, such as the ACSM measurement time resolution, the detection limits for NR species.*

We have added a sentence in the line 140, P 7 of Section 2.2 (revised version): "*In this study, we recorded the data with a time resolution of 30 minutes.*" We did not conduct a detection limit analysis in this study. However,the mass closure analysis was described in the second paragraph of this section. Also, Section 1 of the SI described the quality of the CE corrected data using scatter plots (Fig. S1) vs collocated independent measurements.

*Line 148, change "doesn't" to "don't"*

We have moved this part into Section 1 of the SI, but of course, the typo has been corrected.

*Line 179, what's the reason for the ion signals at m/z 12 and 13 being mostly negative? Is this an issue specific to this study?*

It is not unusual for *m/z* 12 and 13 to be problematic in Q-ACSM data. To our knowledge, a conclusive reason for this has not been reported, but we speculate it may be due to electronic instability at the beginning of the quadrupole scan. However, their concentration is in any case quite low, and their exclusion from PMF therefore does not affect the results.

*Line 179 -180, why is m/z 15 "affected by high biases due to potential interference with air signals"? ACSM determines particle signals as the difference between the filter-off and filter-on modes. So, aren't the air influences on the ACSM diff signals removed from the diff signals?*

Some organic ions still need to correct some air interferences even after the subtraction of sample and filter signals, *m/z* 15 is one of them. In general, ACSM/AMS uses the fragmentation table created by Allan et al. (2004). The *m/z* 15 from the organic matrix is calculated using the following equations:

frag_organic[15]=[15]-frag_NH$_4$[15]-frag_air[15]

frag_air[15]=0.00368*frag_air[14]

frag_air[14]=[14]-frag_nitrate[14]

When we refer to the potential air interference, the coefficient (0.00368) we used to estimate frag_air[15] could vary from different environments because of the different proportions of the N15 isotope in the air in different environments. A similar phenomenon could also be observed for *m/z* 29 organic signal due to a similar issue. However, the lack of resolution in *m/z* of the ACSM makes it difficult to perfectly isolate the air interference. Therefore, we have to remove it before PMF analysis.

*Line 251, the meaning of "within a range of 0.4" is vague. Spell out the range. What exactly does "random" mean in "a random a-value"?*

We appreciate the input from the reviewer. We have moved this section (PMF Window settings) into the Section 3.2.3 of the SI, and the sentence has been changed to: *"To allow the factor profile to adapt itself over time, we applied an a-value randomly from a set of a-values, including 0, 0.1, 0.2, 0.3, and 0.4 (so-called random a-value approach)."* Here, "random" means that with 50 repeats per PMF window, the PMF window can apply *a*-values randomly within the range of 0-0.4 (Δ*a*-value=0.1).

*Section "2.6.1 Window settings", many discussions within this section do not seem closely related to the topic of setting the proper rolling window size.*

We have changed the section name to "PMF window settings". Here, we mention that we test the optimum window size using different window lengths (1, 7, 14, and 28 days) in this section. But of course, we do not mention the optimum window size yet, since it would be covered in Section 4 of SI.

*Line 254- 255, it is mentioned that different a-values were chosen to constrain the fitting of LOA. is there any significance with the specific value?*

Not really, we can get this 58-OA even from unconstrained PMF. The changes of different *a*-values do not have significant influence on the PMF solutions. However, since the factor profile is always stable and unique, we did not attempt the random a-value approach during rolling analysis to save some computational time. Specifically, let's say we had an upper a-value for 58-OA of 0.4 with the Δ*a*-value of 0.1. Then, it would have 5 possible a-values for 58-OA. When we consider HOA and BBOA in this study, we will end up with 5*5*5=125 possibilities. In order to cover all the possibilities during the bootstrap process, we have to set the iterations of each time window to more than 125. In the end, the total PMF runs will be 2.5 times more than the current PMF runs. Therefore, we did not consider applying a random *a*-value approach for this factor.

*Line 257, does this sentence suggest surface ionisation enhances the production of N-containing ions? Are there reference(s) to support this?*

Yes, we do suggest that. But we can only speculate on the potential chemical fingerprints of these three ions ($m/z$ 58, 84, and 98) based on the NIST database. But we cannot make a strong statement with the poor $m/z$ resolution of the ACSM.

*As the LOA factor appeared after a filament change, it is a factor associated with a certain instrument condition. ME-2 analysis that constraints the LOA profile may lead to the retrieval of this factor may be forced even when it was not supposed to be present.*

The 58-OA is always present for the spring, summer, and fall 2014 seasons during unconstrained PMF. Therefore, we do believe the presence of it after the filament change. Regarding the fall 2013 and winter seasons, it is true that we cannot identify 58-OA from unconstrained PMF, but the contribution of this factor is so small (1.2% and 0.7% in Fig. 4) that we think it is OK to leave it in for the yearlong rolling analysis. We do not want to run rolling PMF separately with a different number of factors to avoid an inconsistent transition period. Therefore, we decided to keep this factor for the whole year although its contribution before the filament change was negligible.

*Line 277, what is eBCtr ?*

Thanks a lot for the notice, we have added the descriptions for this acronym and eBCwb at the place where they first occur.

*Line 282, isn't the measurements done with Q-AMS, how was it known that m/z 43 is C2H3O+?*

We removed (Canonaco et al., 2015) from the citation. Ng et al. (2010) used Q-AMS datasets for the analysis, and they did conclude that $m/z$ 43 of OOA factors is mainly from $C_2H_3O^+$. In addition, we added (Crippa et al., 2014) to the citation. This overview study includes quite a bit of HR-AMS measurements, they also clearly stated that $m/z$ 43 of OOA factors is mainly from $C_2H_3O^+$.

*Line 287 – 288, what do 4th and 5th position reference to?*

In total, we have 5 facotrs with two unconstrained factors (MO-OOA and LO-OOA) at either $4^{th}$ or $5^{th}$ position in all 20750 PMF runs.Therefore, we need to put the MO-OOA factor into the same position ($4^{th}$ position in this study) for all 20750 runs before averaging. But of course, we need to explain it better. Therefore, the sentence has been rephrased to: *"Since we left two factors unconstrained ($4^{th}$ and $5^{th}$ factor), MO-OOA can be at either at the $4^{th}$ or the $5^{th}$ position in these 20750 runs."*

*Line 316, define BLH*

Modified.

*Line 325-328, this sentence is awkward and difficult to understand. Consider to revise.*

We thank the suggestion from the reviewer, we have modified it to: *"It was due to advection within the shallow boundary layer as both primary and secondary pollutants increased simultaneously. At the same time, the local wind speed near the ground was very low. One potential explanation was that the locally and regionally induced orography influenced winds, including vertical diffusion processes, caused these delayed midday peaks. However, these processes remain difficult to track without spatially distributed measurements."*

*Line 398, the small seasonal differences in HOA% is interesting. What's the explanation? Since HOA is mainly a POA factor whose concentration should be influenced strongly by BLH, it loading tends to be much higher during winter than in summer. In contrast, stronger photochemistry tends for lead to higher SOA in the summer. So the seasonal difference in HOA% is expected to be strong in winter.*

It is true that BLH heavily influences the seasonal variabilities of HOA mass concentration. Also, it is true that the HOA mass concentration is higher in winter (0.74 µg·m$^{-3}$) than in summer (0.57 µg·m$^{-3}$) because of the higher OA loadings in the winter, which was due to the fact that the lower temperature in winter favors components partitioning into the particle phase and and also higher biomass burning related OA sources (BBOA and LO-OOA in winter). However, we cannot state that the HOA contribution was higher in the winter when both HOA and OA mass concentration increased.

*Line 428, what are the standard deviations for the a values?*

The standard deviations have been added in the sentence: *"HOA and BBOA have averaged a-values of 0.207±0.036 and 0.195±0.050, respectively."*

*Line 483, is dimethyl disulphide sufficiently low volatility to be in the particle phase? Are there HR-AMS or other analytical results to support the presence of this compound? As pesticide application is usually seasonal, did you see evidence from this perspective?*

Thanks to the comment from the reviewer. We modified the sentence to *"In July, a potential source of these distinct ions was some oxidation products of dimethyl disulphide, which shows signals at m/z 94, m/z 95, and m/z 96 (NIST Mass Spectrometry Data Center, 2014)."* Despite the high volatility of dimethyl disulphide, considering the monitoring station is literally in the middle of a farmland, we still believe there were possibilities that oxidation products of it could be detected by our online instrument. However, again, this can only be speculated with the poor *m/z* resolution of the ACSM. Nevertheless, it is a potential explanation of this event.

*Line 519 – 521, are there references to support this statement?*

We have changed the statement to:*" This was most likely due to the increase of biogenic sources and the increasing photochemistry (high O$_3$ and NO$_2$ concentration) at high temperature (>20 °C), which caused the complexity of the OOA sources."* These high-temperature points were mainly from summer, and the O$_3$ concentration was also very high at the same time (Figure. R1 (b)). Therefore, we changed the statement by explained it in a better way as the reviewer suggested.

[Figure]

**Figure. R 1** Absolute statistical uncertainties of PMF for Total OOA (LO-OOA+MO-OOA) for all data: (a) The data points are colour-coded by date and time and (b) the data points are colour-coded by O$_3$ concentration (ppb).

*Fig 3, what do BCtr, BCwb stand for?*

These are equivalent black carbon from traffic source (eBCtr), equivalent black carbon from wood burning source (eBCwb), we have added a spelling out the description in the main text. Moreover, we make sure all these terms, eBC, eBCtr, and eBCwb are consistent.

*Fig. S6, explain the error bars in the caption*

Modified to:**"*Fig. S4*** *Averaged factor profiles from seasonal bootstrap solutions for five different periods. The error bars of each factor represent the standard deviation of the averaged bootstrapped solution, the thick dark sticks are the variabilities that each variable allowed to vary with the corresponding averaged a-value. SON = September, October and November, DJF = December, January and February, MAM = March, April, and May, JJA = June, July, and August.*"

*Fig S8, how exact was the probabilities calculated?*

Each PMF time window has 50 repeats, while only part of these 50 repeats would be selected. The probabilities were calculated using the employed *a*-values from selected PMF runs for each time window.

*Fig. S9, the key is difficult to understand, what are the dots exactly, what does it mean "clouds of measured f44 vs. f43 in SOA factors"?*

We added Eq. S11 and Eq. S12 into the SI to describe how these small dots are calculated.

$$subtracted\ f_{44} = \frac{mass\ conc.of\ OOA\ @[m/z\ 44]}{mass\ conc.of\ OOA + residual\ of\ total\ OA} \tag{11}$$

$$subtracted\ f_{43} = \frac{mass\ conc.of\ OOA\ @[m/z\ 43]}{mass\ conc.of\ OOA + residual\ of\ total\ OA} \tag{12}$$

*Fig. S10, figure caption hard to understand. What are the "Missing time points"?*

Modified to: "*Fig. S10*** *Non-modelled time points (due to criteria-based selection) and Q/Q$_{exp.}$ vs rolling window size.*"

**Reference**

Allan, J. D., Delia, A. E., Coe, H., Bower, K. N., Alfarra, M. R. R., Jimenez, J. L., Middlebrook, A. M., Drewnick, F., Onasch, T. B., Canagaratna, M. R., Jayne, J. T., & Worsnop, D. R. (2004). A generalised method for the extraction of chemically resolved mass spectra from Aerodyne aerosol mass spectrometer data. *Journal of Aerosol Science*, *35*(7), 909–922. https://doi.org/10.1016/j.jaerosci.2004.02.007

Canonaco, F., Slowik, J. G., Baltensperger, U., & Prévôt, A. S. H. (2015). Seasonal differences in oxygenated organic aerosol composition: implications for emissions sources and factor analysis. *Atmos. Chem. Phys.*, *15*(12), 6993–7002. https://doi.org/10.5194/acp-15-6993-2015

Crippa, M., Canonaco, F., Lanz, V. A., Äijälä, M., Allan, J. D., Carbone, S., Capes, G., Ceburnis, D., Dall'Osto, M., Day, D. A., DeCarlo, P. F., Ehn, M., Eriksson, A., Freney, E., Hildebrandt Ruiz, L., Hillamo, R., Jimenez, J. L., Junninen, H., Kiendler-Scharr, A., … Prévôt, A. S. H. H. (2014). Organic aerosol components derived from 25 AMS data sets across Europe using a consistent ME-2 based source apportionment approach. *Atmospheric Chemistry and Physics*, *14*(12), 6159–6176. https://doi.org/10.5194/acp-14-6159-2014

Hildebrandt, L., Kostenidou, E., Lanz, V. A., Prevot, A. S. H. H., Baltensperger, U., Mihalopoulos, N., Laaksonen, A., Donahue, N. M., & Pandis, S. N. (2011). Sources and atmospheric processing of organic aerosol in the Mediterranean: insights from aerosol mass spectrometer factor analysis. *Atmospheric Chemistry and Physics*, *11*(23), 12499–12515. https://doi.org/10.5194/acp-11-12499-2011

Ng, N. L., Canagaratna, M. R., Zhang, Q., Jimenez, J. L., Tian, J., Ulbrich, I. M., Kroll, J. H., Docherty, K. S., Chhabra, P. S., Bahreini, R., Murphy, S. M., Seinfeld, J. H., Hildebrandt, L., Donahue, N. M., DeCarlo, P. F., Lanz, V. A., Prévôt, A. S. H. H., Dinar, E., Rudich, Y., & Worsnop, D. R. (2010). Organic aerosol components observed in Northern Hemispheric datasets from Aerosol Mass Spectrometry. *Atmospheric Chemistry and Physics*, *10*(10), 4625–4641. https://doi.org/10.5194/acp-10-4625-2010

NIST Mass Spectrometry Data Center. (2014). Disulfide, dimethyl. In *NIST Chemistry WebBook* (SRD 69). https://webbook.nist.gov/cgi/cbook.cgi?ID=C624920&Mask=200#Refs

---

## Author Comment (AC2) · 19 Jul 2021

We thank all the constructive comments from two reviewers. The following texts are the response to the reviewers.

The normal italic font is original *reviewer comments,* smaller green font is authors' responses, and the small blue italic font is *the changes in the revised version.*

**Anonymous Referee #1**

*The manuscript by Chen et al. describes the Organic Aerosol (OA) Source Apportionment results of one year long measurements carried out by an aerosol chemical speciation monitor (ACSM) in Magadino, a rural village located in the south of the Swiss Alpine region, one of the most polluted areas in Switzerland. The Authors applied positive matrix factorisation (PMF) and identified well known OA factors: two primary OA factors (traffic-related hydrocarbon-like OA, HOA and biomass burning OA, BBOA), and two secondary factors (a less oxidised oxygenated OA, LO-OOA factor, and a more oxidised oxygenated OA, MO-OOA factor) plus a socalled local OA (LOA) factor. The main novelty of the study is the application (for the first time on a rural site) of a new "rolling" algorithm to account for the temporal changes of the source profiles in PMF. Thus, in addition to the description of the main components that characterise OA in the site under examination, the manuscript presents a detailed comparison between the more traditional methods of applying PMF and the new "rolling" method suggested as the best approach to improve the representativeness of the factors identified. The Authors also calculated the uncertainties (σ) for the modelled OA factors (i.e., rotational uncertainty and statistical variability of the sources) finding values ranging from a minimum of ±4% (LOA) to a maximum of ±40% (LO-OOA). The manuscript represents a huge effort with the aim of improving OA source apportionment PMF capability and it is of possible interest for a large audience of the atmospheric organic aerosol community. However, the text results quite hard to read and it is not completely clear in some parts because of a low-quality presentation and organisation of figures and sections (both in the main text and in the SI). Moreover, the significance of the improvement provided by the new method should be clarified better (given the huge amount of analyses required to apply it). For this reason, my suggestion is to accept the paper only after a strong re-organisation of the text and after the consideration of some major issues listed below.*

We thank the reviewer for the positive feedback about the novelties as well as the significances of this study. We also believe this work will be the role model for future studies that want to apply the same approach for a long-term dataset.

*Major issues and general comments:*

*The amount of data presented, the number of different methodologies and settings together with their evaluations make already complex to follow the discussion and understand the main results. Additionally the text is fragmentary, with too many mixing up with SI (sometimes with wrong references). All this makes too hard to follow the discussion and to find out the scientific relevance of the whole work. I strongly recommend to reorganise the paper, summarising in the main text the most fundamental approaches and results and leaving technicalities to the SI (which should be better organised as well).*

We thank the reviewer's valuable suggestions. We have reorganised the structure of the manuscript. We moved most of the methodologies and settings into SI as suggested, and we have added a table of content for the SI to give the reader a good overview of the paper, sequences of the figures in SI were also reordered to follow the storyline for both main text and SI.

*The rolling analysis is for sure an interesting attempt to deal with the temporal changes of the source profiles in PMF running over long-term datasets. On the other hand, it requires a huge amount of PMF runs (several thousands) which are very time consuming and expensive in term of calculation resources. This can be particularly true if someone hypothesise to extend the same approach to high-resolution AMS data or other datasets in which the number of variables can be substantially higher. Moreover, it is not clearly quantified the improvement of using the rolling approach with respect to the seasonal: I mean, if the the rolling is leading to estimations with uncertainties up to 40% for OOA factors and the improvement in quantification of the same factors is of the order of 5-10% someone can argue that the game is not worth of the effort. I'm not saying this is true, but in my opinion the Authors should add a more comprehensive (even if synthetic) description (motivated by numbers) of the advantages/disadvantages of the rolling approach together with recommendations on how and when the approach can be profitably used or not.*

We appreciate the comments from the reviewer. Indeed, rolling PMF requires huge efforts from the user. However, for several year-long datasets, rolling PMF is a faster and easier way to understand the trends of OA sources. In addition, the rolling PMF tends to get two well-separated OOA factors with the smaller rolling window. However, with the seasonal PMF it is sometimes difficult to get a clean separation of two OOA factors, especially during winter seasons. Moreover, compared with conventional seasonal PMF, bootstrap is often necessary to get stable solutions, which requires at least 100 runs per season. For instance, as the winter season has 3000 data points, 100 runs with bootstrap enabled will easily take 4.5 hours (160 secs/run) approximately. However, for the rolling PMF, with 50 repeats per window (window size=14 days), 3000 data points in winter typically need 4500 runs, but it only takes roughly 3 hours (2-3 secs/run). Therefore, rolling PMF is actually computationally less expensive than the conventional seasonal PMF.

We added recommendations on when and how the rolling PMF can be advantageous as well as the limitations of the rolling analysis at the second last paragraph of the Conclusion: "*Therefore, the rolling PMF is highly useful when the user wishes to better separate OOA factors (especially during cold seasons) and better represent the measurements. In addition, we will also recommend using the rolling PMF to facilitate the analysis of long-term trends of OA sources with some prior knowledge of OA sources. However, it remains challenging to objectively define the transition point to an improved source apportionment for rolling PMF analysis when a different number of OA factors is necessary for different periods. An upcoming manuscript Via et al. (in prep.) will present more details of the comparison between rolling and seasonal results for multiple datasets.*"

*Specific comments*

*Abstract P2, L22: "cite" probably misspelled for "city"*

Corrected

*P2, L27-28: the sentence is redundant: it was already introduced few lines above the distinction of OOAs in two main types.*

The sentences have been corrected to *"OOA (sum of LO-OOA and MO-OOA) contributed 71.6% of the OA mass, varying from 62.5% (in winter) to 78% (in spring and summer)."*

*Introduction P4, L83: here as well as along all the text, please revise the references: you should reorder them based on the years of publication.*

Corrected

*P7, L144-146: where is the comparison? Reading this sentence, it seems in Fig. S1 we should find the comparison between different CE, but it's not the case. Please rephrase the sentence clarifying better what is presented in Fig. S1. Consider also that Fig. S1b is not introduced at all in the text neither explained in SI.*

We made the comparison between two different approaches (time-dependent CE and constant CE) in terms of the correlation between these independent measurements (anion mass concentrations from $PM_{2.5}$ filter

samples, chromatography samples, and TEOM data of PM$_{2.5}$ and PM$_{10}$). However, it would be too much to show the comparisons. Therefore, we move this part into the Section 1 in the SI to describe in more detail how we decide on the CE. In addition, the descriptions of Fig. S1b and Fig. S1c have been covered in this section.

*P11, L230 and all Section 2.6: there is a lot of confusion between this section and the Supplementary Section 2. It is difficult to follow and understand the steps and what is happening. For example, you mention here the socalled Local OA factor (LOA) but you introduce it much later in the text (and, since it is not a standard factor, it is difficult to understand the motivation of the described procedure). Moreover, you discuss figures of SI starting from Fig S6 here, and only later the previous ones. For this reason, please consider to reorganise substantially the text between this section and the corresponding SI section.*

We thank the reviewer for the great suggestion. We have moved and reorganised the structure of the paper substantially. In the revised version, we described the major differences and similarities of methodology between this study and Canonaco et al. (2021) in the main text (now last two paragraphs in Section 2.5). Also, the LOA (which has been changed to 58-OA in the revised version) was introduced in both the main text and the SI in a more organized sequence in the revised version. In addition, we moved the detailed descriptions of the steps of our analysis into the SI. Also, we have reorganised the sequences of figures in the SI to follow the storyline for both the main text and the SI.

*P16, L356: again, LOA factor is discussed but it is not introduced/described yet. Please introduce it before commenting on it: since it is not a standard factor you would need to add spectra and explanations before.*

Thanks to the reviewer's suggestion, we discussed the m/z 58 related OA (58-OA) factor in Section 2.5 (revised version), which is before this section.

*P17, L359-364: which kind of tests? It is not clear what you have done here. Consider to rephrase or to remove. Moreover, are you sure you are citing the right figure here? Fig S6 looks unsuitable to me here.*

We thank the reviewer for the suggestion, and we have removed the citing of the figure. It was a test about PMF runs with a different number of factors. We rephrased the sentence to "*For the factor(s) with a secondary origin, we performed PMF models with a different number of factors (3–6) to assess if the oxygenated OA (OOA) factor is separable without mixing with primary organic aerosol (POA) factors (with a high contribution of m/z 44 that is likely dominated by the CO$_2^+$ ion, derived from the decomposition of carboxylic acids (Duplissy et al., 2011). We conducted these tests (with a different number of factors) independently for different seasons (autumn 2013, winter, spring, summer, autumn 2014).*"

*P17-18, L380-391: Consider to anticipate this introduction of LOA factor, as mentioned before. Moreover, why do you call this factor as "Local OA"? If I understood well it is clearly the result of an instrumental artifact: it is interesting that PMF can isolate also this kind of problems, but you should explain and put the emphasis on this.*

As addressed in the previous response, we agree with the reviewer that there is insufficient evidence to conclusively assign this factor to local emissions. Therefore, we now label this factor in terms of its observed spectral features as "58-OA". The surface ionization changes due to the switch of the filament in the middle of the campaign and constraining of this factor during PMF analysis led to overestimation of its mass concentration. The time series of it should be always considered as the upper limit of this source, and the real mass concentration of it could be substantially lower. However, since the mass concentration remains low, we still consider it as a minor factor. Therefore, we only include it in the PMF analysis without further exploration of its potential source. To better explan the 58-OA, we therefore modified the introduction of this factor in Section 2.5 (revised version) as follows: *The 58-OA was dominated by nitrogen-containing fragments (at m/z 58, 84, and 98). In general, ACSM estimates organic m/z 98 signal by dividing organic m/z 84 to a factor of 2 according to the fragmentation table of organic species that was provided by Allan et al. (2004). Thus, the intensity of m/z 98 is always half of the intensity of m/z 84 in each factor. This 58-OA only appeared*

*after the filament was switched on 14 April 2014. The instrument setup thus strongly influenced the sensitivity of these components (likely due to influences of surface ionizsation). The nitrogen-containing ion, m/z 58, was also observed in Hildebrandt et al. (2011) due to the enhanced surface ionisation in a certain period. In addition, the potassium signal enhanced at the same time, which further corroborated our hypothesis of the enhanced surface ionisation. Also, since this factor was constrained through the whole dataset, the PMF model overestimated the mass concentration of this factor significantly, which leads to high uncertainties for the 58-OA. Therefore, the time series of this source should be considered as the upper limit, and the real mass concentration of it could be substantially lower. However, with the low mass concentration of the 58-OA during the whole campaign, we considered it as a minor factor. Thus, this factor was considered in the PMF analysis, but no further interpretation of its potential source will be attempted in this manuscript."*

The PMF model is capable of isolating this kind of factors, indeed, based on results from unconstrained PMF and constrained runs with a different number of factors, we could always identify this factor. This is because both the factor profiles and time series are so distinct that PMF could easily pick it up. Therefore, as the reviewer suggested, we emphasised it by adding the following sentence in this paragraph (P15, line 334 in the clean revised version): "*In addition, the time series and factor profile of 58-OA were so distinct that PMF could easily resolve it.*"

*P18, L394: Supplementary Section 4 is introduced and discussed before Section 3. The Supplementary Figure (Fig. S10) discussed here is instead postponed. This is misleading and create confusion.*

We thank the reviewer for the great suggestion. We have introduced Section 3 before this sentence in the revised version.

*P19, L422-424: why do you introduce Fig. 7 before Fig. 6? This sentence is anticipating the topic creating confusion. Please rephrase or remove it.*

Removed

*P20, L436-437: Could you please elaborate more on the peaking concentrations at 10–11 a.m.? Why the daily maximum is so late in the morning? I'm used to thinking that the diurnal evolution of PBL leads to more dilution and so lower concentrations of pollutants and that it starts earlier in the morning. I suppose there is a late sunlight illumination of the ground at the site, but it should be explained clearly because it sounds weird.*

As discussed in the last paragraph of Section 3.1, the delayed peak concentrations at 10–11 a.m. were most likely due to some meteorological conditions (low wind speed and delayed illumination of the valley site). These meteorological conditions could influence vertical diffusion processes locally and regionally, which caused the delayed peak of all pollutants. We, therefore, modified the sentence to "*As discussed above, during winter, all of the air pollutants, including all PMF factors peaked concurrently at 10–11 a.m. (local time) due to delayed illumination of the valley site and slow wind speed near the ground (light blue markers in Fig. 2 for total PM$_{1.}$ and Fig. 5b).*"

*P23, L519-521: what do you mean with "more complex aging processes"? It is not clear and/or highly speculative. It is actually demonstrated by an increasing number of studies that OOA formation and ageing is complex also under low temperature and dark conditions. If you mean that the higher variability at higher temperature could reflect different sources/precursors emissions enhanced at higher temperature (e.g., higher emissions of biogenic VOC and the subsequent mixing with preexisting OOAs or whatever) you should explain it better.*

We thank the reviewer for this comment. The sentence has been modified to "*This was most likely due to the increase of biogenic emissions and the increasing photochemistry (high O$_3$. and NO$_2$. concentration) at high temperature (>20 °C), which caused the complexity of the OOA sources.*" These high-temperature points

were mainly from summer, and the O$_3$ concentration was also very high at the same time (as shown in **Figure. R 1**). Therefore, we changed the statement to explain it in a better way, as the reviewer suggested.

[Figure]

**Figure. R 1** Absolute statistical uncertainties of PMF for Total OOA (LO-OOA+MO-OOA) for all data: (a) The data points are colour-coded by date and time and (b) the data points are colour-coded by O$_3$ concentration (ppb).

*P24, L528-534: this is interesting. But the sentence is quite problematic. Can you elaborate more on it? Do you believe it can be also a question of selective water solubility of components?*

The text has been modified into "*Figure 9a shows that the rolling results had a higher OOA concentration during the winter season than the offline PM$_{2.5}$/PM$_{10}$ results, while the rolling results present a lower BBOA concentration during the winter season than the offline PM$_{2.5}$/PM$_{10}$ results (Fig. S11b). As shown in Fig. 9b, the LO-OOA in the rolling results were heavily affected by biomass burning with apparent biomass trace ions (i.e., m/z 60 and 73). The offline results apportioned this biomass burning-affected LO-OOA to BBOA, whereas the online ACSM measurements with a higher time resolution were capable of capturing the fast oxidation process of biomass burning sources. In addition, the rolling PMF technique enabled the LO-OOA factor profile to adapt to the temporal viabilities of OA sources, so the relatively aged biomass burning related OA was apportioned to LO-OOA during winter time by rolling PMF. Therefore, the offline AMS technique tended to underestimate OOA but overestimate BBOA in this study.* "

Daellenbach et al. (2016) suggested a good recovery rates for BBOA and OOA from offline PMF techniques, with 65% and 89%, respectively. In general, OOA is relatively more soluble than BBOA. Of course, the water solubilities of different factors could play a role in this phenomenon, but it was not the main reason behind it. Because the relatively low solubility of BBOA could not explain the offline results which had a higher BBOA concentration, it could not explain the sudden jump of the difference of the OOA concentration during the winter season between the rolling and offline results. Therefore, we do not believe that the selective water solubility of components plays a vital role in this phenomenon.

*Conclusions: expressions like "a somewhat better solution" or "more realistic results" are quite subjective and vague. "More realistic" based on what? What is the real improvement of using rolling PMF instead of more traditional (and less time-consuming) methodologies in term of identification and especially quantification of OA sources? You should clarify it better here in the conclusions. And possibly you should add a more comprehensive description (motivated as much as possible by numbers) of the advantages/disadvantages of the rolling approach together with recommendations on how and when the approach can be profitably used or not.*

Thanks to the reviewer's comments, we rephrased the sentence as follows: "*Overall, the rolling PMF provided slightly better correlations with external tracers, especially between the OOA factors and corresponding inorganic secondary salts. In addition, the rolling PMF results provided a better representation of the measurements by adapting the temporal variations of OOA factors in the f$_{44}$ vs f$_{43}$ space, which also led to*

*much smaller scaled residuals than for the seasonal PMF. Therefore, the rolling PMF is highly useful when the user wishes to better separate OOA factors (especially during cold seasons) and better represent the measurements. In addition, we will also recommend using the rolling PMF to facilitate the analysis of long-term trends of OA sources with some prior knowledge of OA sources. However, it remains challenging to objectively define the transition point to an improved source apportionment for rolling PMF analysis when a different number of OA factors is necessary for different periods."*

*Figures General low quality, with too small font sizes making difficult to read labels of the axes and legends. Sometimes problematic also the choice of colors (e.g., Fig. 5 and 9a). Please check the readability of all the figures in the main text, in the Appendix A and in SI.*

We really appreciated the reviewer's suggestions. We have improved all figures as recommended. Please find more details in the revised version.

*Figure 3: given that the x-axis (the time period) is common to all the panels, please consider to use only one or at least to make them of the same length (in order to improve the readability and the comparison between the different time trends)*

Modified as suggested:

[Figure]

**Fig. 3** Annual cycles of OA components: (a) absolute and (b) relative OA contributions plotted as 30-min resolved time series, (c) BC source apportionment.

*Figure 5: all the labels (axes name and values, legend, etc.) are difficult to read. Especially graphs in panel b are completely not aligned, their legend is unreadable and the colors of the time series are misleading. Please increase the font size of all legends and labels and improve the general format of the figure.*

Modified as suggested:

[Figure]

**Fig. 5** Overview of the primary and secondary OA components in Magadino in 2013-2014: (a) OA factor profiles and (b) seasonal diurnal cycles of HOA, BBOA, LOA, MO-OOA, and LO-OOA. The ambient temperature is shown on the LO-OOA diurnal plots. In (a) the error bar is the standard deviation; the black bars show the maximum and the minimum that the variable was allowed to vary from the reference profiles.

The average, 10th, and 90th percentiles for a-values of HOA are 0.195, 0.007 and 0.378, respectively. Also, the average, 10th, and 90th percentiles for a-values of BBOA are 0.202, 0.025 and 0.379, respectively.

*Supporting Information Check if the Figures follow the order of presentation in the text: Figure S6 should be anticipated (because discussed before, at the beginning of Section S2). In order to improve the readability it is also important to put intervals and possibly titles between subsections or figures/tables referring to different tests/results/data.*

We appreciate the reviewer's suggestion. We have restructured the SI completely by moving some text into SI from the main text, adding a table of contents, etc. We also reorganise the sequences of figures to follow the storyline of both main text and SI. Therefore the Figure S6 is Figure S4 now in the revised version.

*Fig. S1b is not introduced at all in the main text neither explained here in SI. You need to explain what it is showing (for instance, what is the mini-denuder? Where it is introduced?)*

We added in the first paragraph of the SI the description of Fig. S1b*:" It also showed relatively good consistencies with the anions measured using chromatography from Mini-denuder (MD) (Dämmgen et al., 2010) samples (Fig. S1b)." We added Section 1 of SI to describe all the figures in Fig. S1.*

*Section 2 P5, L62-63: do you have any references for this? What do you mean with "more accurate estimations"?*

We do not have any reference here, but we found it during this study. The statement has been modified to*: "Thus, site-depended reference profiles are necessary (at least for BBOA) to get more accurate estimations of OA sources (better correlation with external tracers in this study when compared to the PMF solution using literature reference profiles)."* More accurate means higher correlation with corresponding external tracers.

*P6, L88-91: it is hard to understand how do you use t-test. This is probably better explained in Canonaco et al. 2020, but it is important to spend some more words also here to improve the understanding of the readers.*

*P6, L96-99: The same comment, it is hard to understand the procedure. Please rephrase and explain better.*

Canonaco et al. (2021) didn't introduce the *t*-test yet. It was the first time to be introduced in this study. But we appreciate this suggestion. In Section 3.3 of the SI (revised version), we rephrased the whole section to better explain the *t*-test as well as the improvements of this technique compared with the 10$^{th}$ percentile technique Canonaco et al. (2021) proposed.

*Fig. S5: labels of the axes are missing. Please add them and increase the font size of the color-legend.*
Modified to:

[Figure]

**Fig. S3** Measured absolute mass concentrations of *mass-to-charge ratio* (*m/z*)=55 and *m/z*=57 with colour coded by hours of the day (a) and date and time (b).

**Reference**

Allan, J. D., Delia, A. E., Coe, H., Bower, K. N., Alfarra, M. R. R., Jimenez, J. L., Middlebrook, A. M., Drewnick, F., Onasch, T. B., Canagaratna, M. R., Jayne, J. T., & Worsnop, D. R. (2004). A generalised method for the extraction of chemically resolved mass spectra from Aerodyne aerosol mass spectrometer data. *Journal of Aerosol Science*, *35*(7), 909–922. https://doi.org/10.1016/j.jaerosci.2004.02.007

Canonaco, F., Tobler, A., Chen, G., Sosedova, Y., Slowik, J. G. G., Bozzetti, C., Daellenbach, K. R., El Haddad, I., Crippa, M., Huang, R.-J., Furger, M., Baltensperger, U., Prévôt, A. S. H., Kaspar Rudolf Haddad, I. E. D., Crippa, M., Huang, R.-J., Furger, M., Baltensperger, U., Prevot, A. S. H., … Prevot, A. S. H. (2021). A new method for long-term source apportionment with time-dependent factor profiles and uncertainty assessment using SoFi Pro: application to 1 year of organic aerosol data. *Atmospheric Measurement Techniques*, *14*(2), 923–943. https://doi.org/10.5194/amt-14-923-2021

Daellenbach, K. R., Bozzetti, C., Křepelová, A., Canonaco, F., Wolf, R., Zotter, P., Fermo, P., Crippa, M., Slowik, J. G., Sosedova, Y., Zhang, Y., Huang, R.-J. J., Poulain, L., Szidat, S., Baltensperger, U., El Haddad, I., & Prévôt, A. S. H. H. (2016). Characterization and source apportionment of organic aerosol using offline aerosol mass spectrometry. *Atmospheric Measurement Techniques*, *9*(1), 23–39. https://doi.org/10.5194/amt-9-23-2016

Dämmgen, U., Thöni, L., Lumpp, R., Gilke, K., Seitler, E., & Bullinger, M. (2010). Feldexperiment zum Methoden- vergleich von Ammoniak- und messungen in der Umgebungsluft , 2005 bis 2008 in Braunschweig. *VTI Agriculture and Forestry Research - Sonderheft*, *337*, 62. https://www.thuenen.de/media/publikationen/landbauforschung-sonderhefte/lbf_sh337.pdf

Duplissy, J., DeCarlo, P. F., Dommen, J., Alfarra, M. R., Metzger, A., Barmpadimos, I., Prevot, A. S. H., Weingartner, E., Tritscher, T., Gysel, M., Aiken, A. C., Jimenez, J. L., Canagaratna, M. R., Worsnop, D. R., Collins, D. R., Tomlinson, J., & Baltensperger, U. (2011). Relating hygroscopicity and composition of organic aerosol particulate matter. *Atmospheric Chemistry and Physics*, *11*(3), 1155–1165. https://doi.org/10.5194/acp-11-1155-2011

Hildebrandt, L., Kostenidou, E., Lanz, V. A., Prevot, A. S. H. H., Baltensperger, U., Mihalopoulos, N., Laaksonen, A., Donahue, N. M., & Pandis, S. N. (2011). Sources and atmospheric processing of organic aerosol in the Mediterranean: insights from aerosol mass spectrometer factor analysis. *Atmospheric Chemistry and Physics*, *11*(23), 12499–12515. https://doi.org/10.5194/acp-11-12499-2011

Via, M. et. al.: *Comparison between rolling and seasonal PMF techniques for organic aerosol source apportionment.* [unpublished manuscript]., 2021